# NMNAT2 is a druggable target to drive neuronal NAD production

James R. Tribble [1,15], Melissa Jöe [1,15], Carmine Varricchio[2], Amin Otmani[1], Alessio Canovai [1,3], Baninia Habchi [4,5,6], Evangelia Daskalakis[4,5], Romanas Chaleckis [4,5,7], Andrea Loreto [8,9], Jonathan Gilley[8], Craig E. Wheelock [4,5], Gauti Jóhannesson[10,11], Raymond C. B. Wong [12,13], Michael P. Coleman [8], Andrea Brancale [2,14] & Pete A. Williams [1] ✉

Maintenance of NAD pools is critical for neuronal survival. The capacity to maintain NAD pools declines in neurodegenerative disease. We identify that low NMNAT2, the critical neuronal NAD producing enzyme, drives retinal susceptibility to neurodegenerative insults. As proof of concept, gene therapy over-expressing full length human NMNAT2 is neuroprotective. To pharmacologically target NMNAT2, we identify that epigallocatechin gallate (EGCG) can drive NAD production in neurons through an NMNAT2 and NMN dependent mechanism. We confirm this by pharmacological and genetic inhibition of the NAD-salvage pathway. EGCG is neuroprotective in rodent (mixed sex) and human models of retinal neurodegeneration. As EGCG has poor drug-like qualities, we use it as a tool compound to generate novel small molecules which drive neuronal NAD production and provide neuroprotection. This class of NMNAT2 targeted small molecules could have an important therapeutic impact for neurodegenerative disease following further drug development.

Neurodegenerative disease is a significant global health and economic burden. NAD homoeostasis is a critical factor that influences neurodegeneration and neuroprotection. Increasing levels of NAD provide neuroprotection in multiple cell and animal models of disease and in human clinical trials[1]. Glaucoma is one of the most prevalent neurodegenerations which affects ~80 million people worldwide[2]. In glaucoma, the progressive dysfunction and loss of retinal ganglion cells (RGCs; the output neuron of the retina whose axons make up the optic nerve) results in irreversible blindness. There are no clinically available neuroprotective strategies.

Recent animal and human studies have uncovered metabolic dysfunction occurring early in RGCs in glaucoma, in particular the critical dependency of RGCs on sufficient levels of NAD[3–5]. In neurons, NAD levels are maintained predominantly through the NAD-salvage pathway's two terminal enzymes; NMNAT1 (localized to the nucleus) and NMNAT2 (localized in the cytoplasm[6]). Protein expression of

[1]Department of Clinical Neuroscience, Division of Eye and Vision, St. Erik Eye Hospital; Karolinska Institutet, Stockholm, Sweden. [2]School of Pharmacy and Pharmaceutical Sciences; Cardiff University, Cardiff, Wales, UK. [3]Department of Biology, University of Pisa, 56127 Pisa, Italy. [4]Unit of Integrative Metabolomics, Institute of Environmental Medicine, Karolinska Institute, Stockholm, Sweden. [5]Department of Respiratory Medicine and Allergy, Karolinska University Hospital, Stockholm, Sweden. [6]C2VN, INRAE, INSERM, Aix Marseille University, 13007 Marseille, France. [7]Gunma Initiative for Advanced Research (GIAR), Gunma University, Maebashi, Japan. [8]John van Geest Centre for Brain Repair, Department of Clinical Neurosciences; University of Cambridge, Cambridge, UK. [9]School of Medical Sciences and Save Sight Institute, Charles Perkins Centre, Faculty of Medicine and Health, The University of Sydney, Sydney, NSW, Australia. [10]Department of Clinical Sciences, Ophthalmology, Umeå University, 901 85 Umeå, Sweden. [11]Wallenberg Centre of Molecular Medicine, Umeå University, 901 85 Umeå, Sweden. [12]Centre for Eye Research Australia, Royal Victorian Eye and Ear Hospital, East Melbourne, Australia. [13]Ophthalmology, Department of Surgery, University of Melbourne, East Melbourne, Victoria, Australia. [14]Vysoká škola chemicko-technologická v Praze, Prague, Czech Republic. [15]These authors contributed equally: James R. Tribble, Melissa Jöe. ✉e-mail: pete.williams@ki.se

NMNAT2 is predominantly neuronal and its NAD-producing activity is essential for survival of long axons[7]. We previously identified down-regulation of NMNAT2 occurring in RGCs prior to neurodegeneration in the DBA/2J mouse model of glaucoma[3,8]. This was subsequently supported by sequencing of translating mRNAs isolated from RGC ribosomes at a degenerative timepoint (where RGC loss has occurred) in a mouse ocular hypertensive (OHT) model[9]. Similarly, we have demonstrated that NMNAT2 immuno-labelling is decreased in late-stage glaucoma in the human retina and optic nerve head (ONH; a critical site of injury to RGC axons), where substantial RGC death has occurred[6].

Given the critical role that NMNAT2 plays in axon maintenance and degeneration, understanding how NMNAT2 levels may influence RGC degeneration is of particular importance. Whilst genetic targeting of NMNAT2 has been demonstrated to be robustly neuroprotective in other neuronal systems, there are no identified drugs or compounds that target endogenous NMNAT2 to produce high levels of NAD in neurons. In this work, we identify that epigallocatechin gallate (EGCG) drives NAD production in neurons through an *NMNAT2*-dependent mechanism. Using EGCG as a tool compound we develop small molecules driving neuronal NAD production through NMNAT2 which can also provide neuroprotection against RGC injury ex vivo.

## Results

### NMNAT2 levels modulate retinal ganglion cell susceptibility to neurodegeneration

NMNAT2 is an essential enzyme for NAD synthesis in neurons. NMNAT2 (RNA and protein) declines in the brain in Alzheimer's disease and its expression is highly variable[10]. To explore NMNAT2 in the retina we first queried publicly available RNA-sequencing and microarray datasets. We demonstrate that across recombinant inbred BXD strains (recombinant inbred lines from crosses between C57BL/6J mice (B6) and DBA/2J mice (D2); to explore *Nmnat2* variations within a non-pathological range), *Nmnat2* expression is highly variable (up to 2-fold difference) among individual strains and is variable within independent datasets for whole eye, retina, and midbrain (where RGC axons terminate; Fig. 1A) suggesting variability in expression within the whole visual system. In the retina, this variability is not related to the number of RGCs (Spearman's rank correlation $r = -0.00088$, $P = 0.997$; Fig. 1B) which is known to vary across individual mice and strains[11]. Supporting this, *Nmnat2* expression has a significantly greater variance than RGC markers *Pou4f1* ($P < 0.001$), *Rbpms* ($P < 0.001$), and *Tubb3* ($P = 0.006$), but not *Thy1* ($P = 0.124$; Fig. 1B). This variability increases with age (Fig. 1C). Likewise, data from human retina demonstrates substantial variability of *NMNAT2* gene expression across individuals (up to 45-fold difference, with 1.8 fold difference across the interquartile range; Fig. 1D). Single-cell and single-nucleus RNA-sequencing from human retina confirms that retinal expression of *NMNAT2* is highly variable across individual RGCs, which have the highest average expression among retinal neurons (Fig. 1E; *NMNAT2* expression is not present in non-neuronal cell types in the retina). We hypothesize that variable expression of *NMNAT2* may be a contributing factor to the heterogeneity of glaucomatous disease.

Age, genetics, and high intraocular pressure (IOP) are all considerable risk factors for glaucoma. In the D2 mouse (a model of a complex age- and IOP- dependent inherited glaucoma), *Nmnat2* expression (as assessed by whole tissue microarray[12,13]) is significantly reduced prior to detectable neurodegeneration in the ONH (molecular disease group 3, $P = 0.025$) and continues to decline in moderate and severe disease (molecular disease group 4, $P = 0.001$, and 5, $P < 0.001$; Fig. 1F). In the retina, *Nmnat2* is significantly reduced in severe disease (molecular disease group 4, $P < 0.001$; Fig. 1F). However, when only considering RGCs (FAC sorted, bulk RNA-sequencing[3]), *Nmnat2* is significantly decreased at a time point prior to detectable neurodegeneration in eyes with the greatest degree of transcriptional

dysfunction (molecular disease group 4, $P = 0.004$; Fig. 1F). Dissecting these glaucoma risk variables, IOP alone drives *Nmnat2* depletion (in the rat bead ocular hypertensive (OHT) model of glaucoma at a time point where IOP is high but there is no significant RGC loss; Fig. 1G and Supp Fig. 1A, B). Direct axotomy of RGCs (severing of the optic nerve and followed by retina tissue culture ex vivo for 12 h) also drives *Nmnat2* depletion in the absence of IOP or age (Fig. 1G). We therefore hypothesized that low NMNAT2 leads to RGC degeneration or renders RGCs susceptible to neurodegenerative insults. To test this, we used mice carrying one or two, fully or semi-penetrant, gene-trapped (gt) alleles for *Nmnat2* to titrate *Nmnat2* levels in the retina (Fig. 2A). In these mice, retinal *Nmnat2* was depleted to (observed *vs.* expected based on allele penetrance) 80% (expected 75%; *Nmnat2*[gtBay/+]), 39% (expected 50%; *Nmnat2*[gtE/+]), or 10% (expected 25%; *Nmnat2*[gtBay/gtE]) of normal levels relative to wild type controls (100%; *Nmnat2*[+/+]). (*NMNAT2* null mice and humans are perinatal lethal[14]). Thus, these mice model the wide spectrum of NMNAT2 levels seen in the human retina (Fig. 1D, E).

Low *Nmnat2* (*Nmnat2*[gtBay/gtE]) results in a developmental or very early post-natal reduction in RGC density followed by an age-related loss of RGCs initiated between 6 and 12 months of age which is absent in *Nmnat2*[+/+] mice up to 22 months of age (Fig. 2B) (although further experiments are required to elucidate the exact timing of this neuronal reduction). This represents an ocular demonstration recapitulating a combination of the developmental loss of peripheral sensory axons and the late onset loss of motor axons previously identified in *Nmnat2*[gtBay/gtE] mice[15]. This further highlights the unique fragility of RGCs across neuronal subtypes. Developmentally regulated RGC death and survival is influenced by connectivity in target brain regions[16,17]. Given that neurons in these brain regions will also have reduced *Nmnat2*, there is likely to be a more limited capacity to maintain RGC connectivity, which may partially explain the developmental and early onset degeneration of RGCs in these mice. Even when accounting for this developmental dropout in RGC populations, RGC loss was significantly greater in *Nmnat2*[gtBay/gtE] mice following axotomy (but not for the single alleles; *i.e.* predicted 75% and 50% *Nmnat2*) (Fig. 2C). This suggests a threshold of *NMNAT2* expression past which RGCs are sensitized to neurodegenerative insults (these thresholds are only met with genetic depletion and are not met within the normal variations within the population). This is supported by the wide range of NMNAT2 levels in humans which are non-pathogenic in the absence of other neurodegenerative insults (e.g. NMNAT2 expression is highly variable in aged postmortem human brains, and further decreased in brains with Alzheimer's disease (~75–100% of controls levels)[10]. Overexpression of full length human *NMNAT2* (hNMNAT2) in RGCs via viral gene therapy was able to overcome RGC susceptibility induced by low genetic levels of *Nmnat2* (*i.e. Nmnat2*[gtBay/gtE] mice; Fig. 3A) and provided complete neuroprotection in B6 mice (where normal physiological levels of *Nmnat2* are present; Fig. 3B). Nmnat2 has been demonstrated to have a neuroprotective role in animal models of neurodegeneration[9,18]. Supporting this hNMNAT2 gene therapy also provided a robust neuroprotective effect in vivo in a rat OHT glaucoma model demonstrating efficacy against the multifactorial insults of glaucoma (Fig. 3C).

### Epigallocatechin gallate drives NAD production through an NMN and NMNAT2-dependent mechanism

Whilst gene therapy has found success in treating monogenic eye diseases, it is less likely to provide complete protection for complex polygenic neurodegenerative diseases such as glaucoma, and its licensed use is likely to face many regulatory hurdles. As such, pharmacological targeting of NMNAT2 may be a viable alternative that will be more amenable in a clinical trial setting. Epigallocatechin gallate (EGCG) is a green tea polyphenol that has previously been identified as a potential NMNAT1-3 activator (in a cell-free enzymatic assay)[19]. We

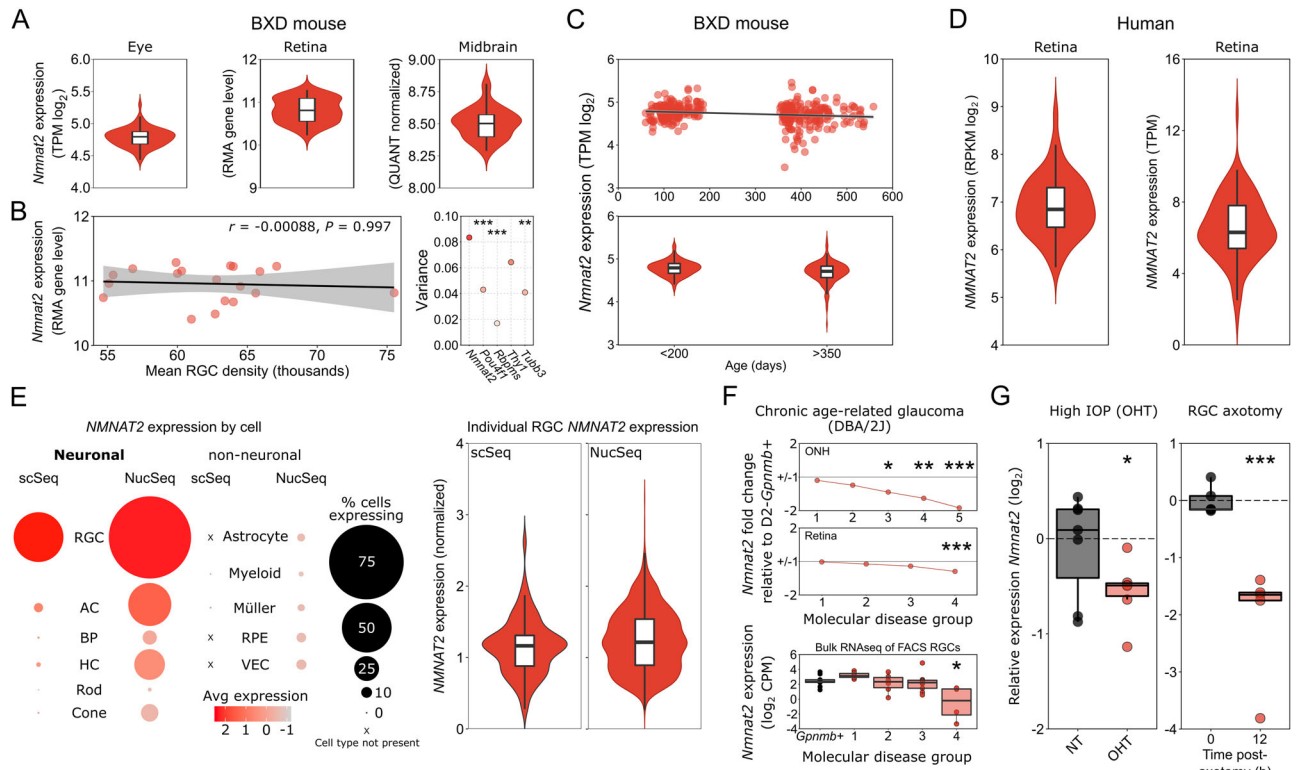

**Fig. 1 | *NMNAT2* expression is highly variable across the visual system tissues and within retinal ganglion cell populations, and declines following injury.**
**A** *Nmnat2* expression is highly variable between individual BXD mice (note logarithmic scales) across RGC relevant tissues (whole eye, $n = 157$; retina, $n = 55$; midbrain, $n = 37$ animals). It is important to note that these data are from different datasets and experiments and therefore should not be used for comparison across tissues, only within tissues. **B** In the retina, this variability is not related to the number of RGCs (*Spearman's* rank correlation, shaded area = 95% CI) and the variance in *Nmnat2* expression is significantly greater than for RGC markers *Pou4f1*, *Rbpms*, and *Tubb3* (*Pitman-Morgan* test of variance for paired samples, two-sided). Both the founder strains (B6 and D2) are represented in these data and neither of these strains have the most upper or lower values (D2 = 10.69, B6 = 11.09, min for series = 10.23, max for series = 11.29 RMA gene level). **C** This variability in *Nmnat2* increases with age (<200 days, $n = 157$ or >350 days, $n = 187$). **D** *NMNAT2* expression in human retina is also highly variable between individuals (left, $n = 50$; right, $n = 105$). **E** Within the cell types of the retina, RGCs demonstrate the greatest average expression of *NMNAT2* by single cell and single nucleus RNA-sequencing (red = highest expression, peach = lowest, dot plot scaled by % of expressing cells within types; AC amacrine cell, BP bipolar cell, HC horizontal cell, RPE retinal pigment epithelium) data from[6] and[47]. Even within these individual RGCs, *NMNAT2* expression is highly variable ($n = 74$ RGCs for scSeq, $n = 2039$ RGCs for NucSeq).

**F** In D2 mice (a chronic, age-related mouse model of glaucoma), *Nmnat2* expression declines in whole ONH at a pre/early-degenerative time point (stage 3) and in retina declines in late disease (stage 4) relative to DBA/2J-*Gpnmb*[RISOX] (for ONH, Group 1 ($n = 8$), Group 2 ($n = 8$), Group 3 ($n = 6$), Group 4 ($n = 4$), Group 5 ($n = 4$) where expression is compared to $n = 5$ D2-Gpnmb + ; in the retina, Group 1 ($n = 8$), Group 2 ($n = 9$), Group 3 ($n = 3$), Group 4 ($n = 10$); expression is compared to $n = 8$ D2-Gpnmb +). In sorted RGCs from D2 retina, this decline in *Nmnat2* is detectable at an early, pre-degenerative time point in RGCs with high RNA dysregulation (Group 1 ($n = 9$ age-matched D2-Gpnmb+ and $n = 6$ D2s), Group 2 ($n = 6$ D2s), Group 3 ($n = 10$ D2s), Group 4 ($n = 4$ D2s). Data in F from[3,8], significance as FDR). **G** This is replicated under the isolated glaucomatous insults of ocular hypertension (whole optic nerves from rat inducible model following 7 days of high IOP (OHT), $n = 6$; and normotensive controls, $n = 7$) and following direct RGC injury through axotomy (whole retina in retinal explant model, 12 h culture ex vivo following axotomy, $n = 5$; or controls, $n = 5$; two-sided *Student's t*-test). For **A**–**C**: TPM transcripts per million, RMA robust multiarray analysis, QUANT quantile, RPKM reads per kilobase of exon per million. Data in **A**–**F** were generated through screening publicly available datasets (*see* Methods). *$P < 0.05$, **$P < 0.01$, ***$P < 0.001$, NS = non-significant ($P > 0.05$). For box plots, the centre hinge represents the median with upper and lower hinges representing the first and third quartiles; whiskers represent 1.5 times the interquartile range.

demonstrate that EGCG, in the nM to µM range, drives NAD production in neurons in a dose (Fig. 4A) and time (Fig. 4B) dependent manner. This effect is not replicated in neuron-low tissues (spleen, muscle, and liver; Fig. 4C). EGCG provides full neuroprotection against RGC loss following axotomy (Fig. 4D) mirroring the *hNMNAT2* gene therapy and our previous work with nicotinamide (NAM; an upstream precursor to NAD in the salvage pathway[3,20]). Further supporting this, EGCG delivered orally reduced (but not fully prevented) glaucomatous neurodegeneration in the rat OHT glaucoma model, with this neuroprotection improving with the addition of *hNMNAT2* gene therapy (Fig. 4E). EGCG also significantly improved RGC survival in human retina punches maintained in culture following postmortem axotomy following enucleation (Fig. 4F). Mitochondrial impairment is a trigger for neurite depletion of NMNAT2, resulting in neurodegeneration[21]. Primary retinal neurons treated with EGCG did not degenerate to the same degree as untreated neurons when exposed to Complex I

inhibition via rotenone (Fig. 5A). EGCG, administered orally, significantly improved RGC survival following in vivo intravitreal rotenone injection which results in rapid RGC loss (Fig. 5B). Cortical neurons demonstrated a significant increase in mitochondria membrane potential (via JC-1 staining) when treated with 5 µM EGCG and a significant reduction in membrane potential at 50 µM (Supp Fig. 2). Whole green tea polyphenols (containing ~25% EGCG by weight) recapitulated the NAD-boosting effects (Supp Fig. 3A) and neuroprotective effects in the retina explant model at doses with comparable EGCG content (Supp Fig. 3B) but did not provide protection in rat OHT (Supp Fig. 3C).

We next determined whether EGCG's neuroprotective effects were NMNAT2-dependent. We depleted NMN (the substrate for NMNAT2) using FK866 (a specific NAMPT inhibitor, the upstream enzyme to NMNAT1 and NMNAT2). When neurons are depleted of NMN, the capacity of EGCG to increase NAD levels is significantly

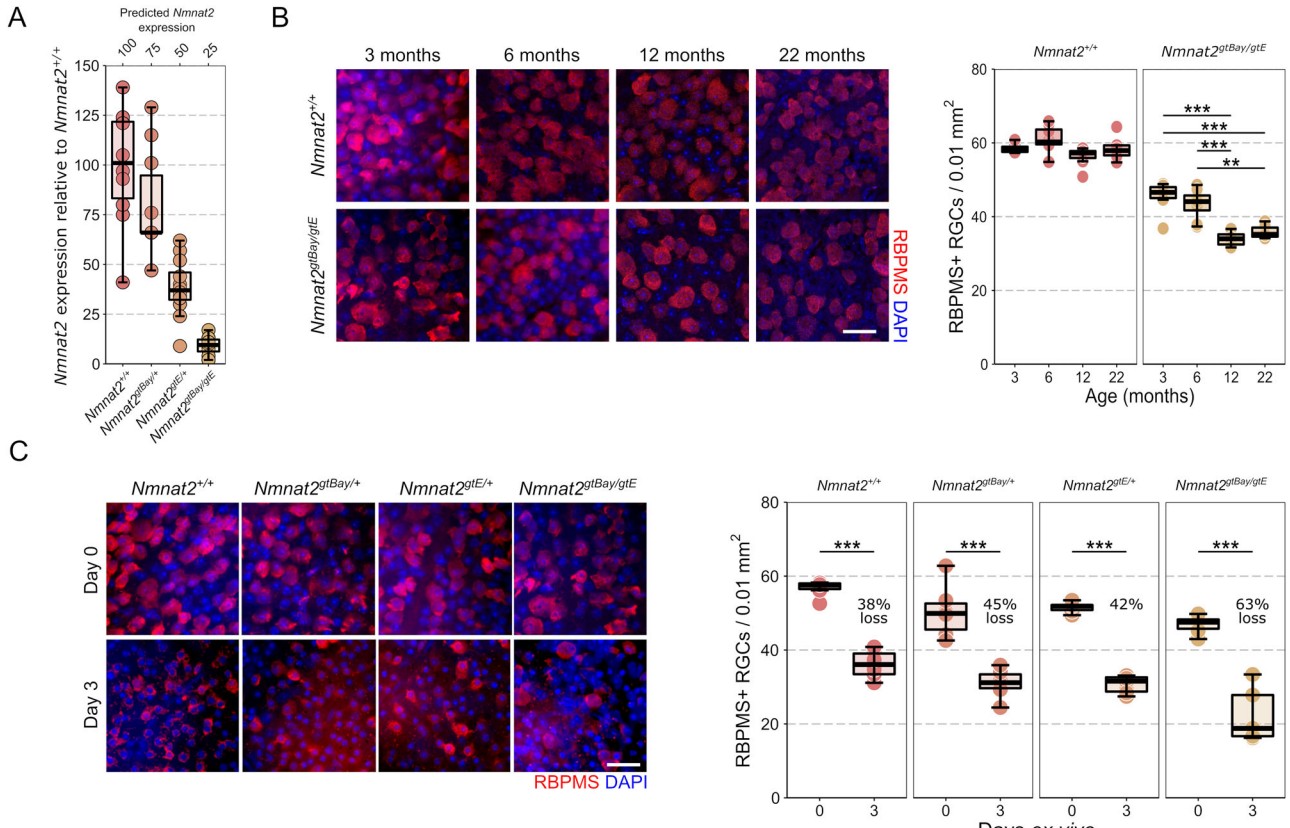

**Fig. 2 | Reduction of *NMNAT2* increases retinal ganglion cell susceptibility to injury. A** Crossing mice heterozygous for *Nmnat2* gene-trap alleles gtBay (predicted 50% silencing) or gtE (predicted 100% silencing) allowed *Nmnat2* titration. In these mice, retinal *Nmnat2* was depleted to (observed *vs.* expected based on allele penetrance) 80% (expected 75%; *Nmnat2^{gtBay/+}*, n = 10), 39% (expected 50%; *Nmnat2^{gtE/+}*, n = 12), or 10% (expected 25%; *Nmnat2^{gtBay/gtE}*, n = 12) of normal levels relative to wild type controls (100%; *Nmnat2^{+/+}*, n = 10). **B** RGC density was significantly lower in *Nmnat2^{gtBay/gtE}* retina than in *Nmnat2^{+/+}* retina at 3 months without further change at 6 months (indicating a developmental loss). By 12 months of age, *Nmnat2^{gtE}* mice had significantly fewer RGCs than at 3 and 6 months, and this is stable to 22 months (indicating an additional early age-related decline). For *Nmnat2^{+/+}*: 3 months, n = 6; 6 months, n = 8; 12 months, n = 8; 22 months, n = 8; for

*Nmnat2^{gtBay/gtE}*: 3 months, n = 6; 6 months, n = 8; 12 months, n = 8; 22 months, n = 6. Scale bar = 20 μm. **C** RGC density was significantly reduced in all *Nmnat2* gene-trap allele mouse strains at 3 days ex vivo following RGC axotomy (RBPMS = specific marker of RGCs in the retina) relative to naïve controls (0 days ex vivo), and this was greatest in *Nmnat2^{gtBay/gtE}* mice supporting a threshold of *Nmnat2* loss beyond which RGC susceptibility to injury is increased (Day 0: *Nmnat2^{+/+}*, n = 6; *Nmnat2^{+/gtBay}*, n = 6; *Nmnat2^{+/gtE}*, n = 6; *Nmnat2^{gtBay/gtE}*, n = 6; Day 3: *Nmnat2^{+/+}*, n = 8; *Nmnat2^{+/gtBay}*, n = 6; *Nmnat2^{+/gtE}*, n = 7; *Nmnat2^{gtBay/gtE}*, n = 5; scale bar = 20 μm). For B and C, *P < 0.05, **P < 0.01, ***P < 0.001, NS = non-significant (P > 0.05); One-way ANOVA with *Tukey's* HSD. For box plots, the centre hinge represents the median with upper and lower hinges representing the first and third quartiles; whiskers represent 1.5 times the interquartile range.

reduced (30% decrease, Fig. 6A) and EGCG no longer protects from neurodegenerative insults (Fig. 6B). This demonstrates that EGCG requires NMN to generate NAD and to provide neuroprotection (*i.e.* an NAD-salvage pathway dependent process requiring NMNAT1 or NMNAT2 to further stimulate NAD production). The neuroprotective effects of EGCG were also reduced in *Nmnat2^{gtBay/gtE}* mice (*i.e.* 25% *Nmnat2* levels), demonstrating an NMNAT-dependent effect (Fig. 6C). Collectively, these data suggested that EGCG may modulate NMNAT2 by binding an allosteric site on the protein surface. However, the crystal structure of NMNAT2 is unknown. To address this and to identify the potential EGCG binding site on NMNAT2, we generated a homology model using the crystal structure of NMNAT1 and NMNAT3 as templates, which was further refined by loop-modelling and extensive molecular dynamic simulations (Fig. 7A, Supp Fig. 4A–C). We identified three consistent potential druggable binding pockets, separate to the catalytic pocket (NMN binding site) on the NMNAT2 surface (Fig. 7B, Supp Fig. 4D, E). In silico, EGCG was able to maintain a stable ligand-protein complex in one of the three pockets identified through hydrophobic contacts and hydrogen bond interactions with the surrounding residues (*i.e.* Gln71, Asp78, Lys151, and Val156) (Fig. 7B, Supplementary Fig. 4F, H) demonstrating the potential for the direct interaction of EGCG with *NMNAT2*.

## EGCG is a tool compound for designing novel, NMNAT2-targeted NAD-boosting small molecules

Although EGCG efficiently drives NAD production in in vitro and ex vivo contexts, and is well tolerated in diet, it is poorly bioavailable and is prone to breakdown at different pHs (as is present throughout the digestive system; Supp Fig. 5)[22]. In addition, EGCG's structural complexity represents a significant synthetic challenge. Together these qualities make EGCG a poor drug for clinical translation. To address this, we screened simplified EGCG-like analogues through a series of step-by-step truncations of the EGCG structure and tested the NAD-boosting capacity in cortical neurons (Group 1, Fig. 7C, D, Supp Fig. 6, Supplementary Data 1). We assessed the effects of the removal/replacement of the hydroxyl group, the influence of stereochemistry, reducing the number of rings, and the replacement of the catechol ring. NAD assays demonstrated that flavone core derivates drastically reduce biological activity while the chromene core partially retains biological activity. This led to the identification of Compound 18 as the simplest structure that showed statistically significant activity, which represented a promising drug-like hit for lead optimization. Analogues of Compound 18 were then tested to explore the structure–activity relationships around the chromene core. The synthetic accessibility of this particular chemical scaffold

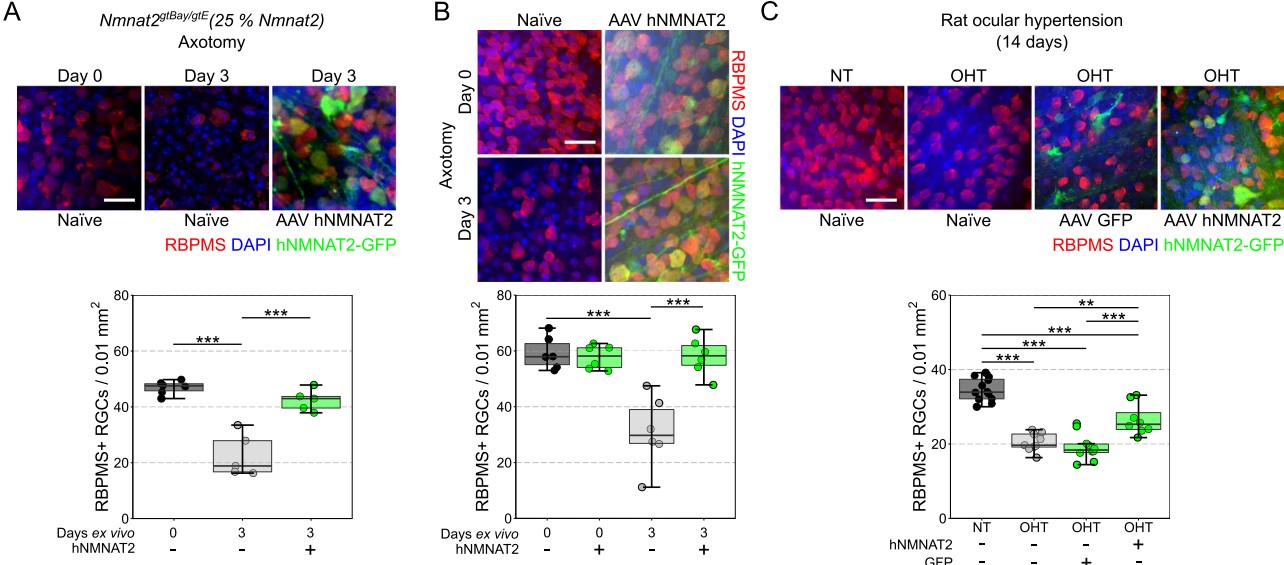

**Fig. 3 | Gene therapy delivery of human NMNAT2 is strongly neuroprotective to retinal ganglion cells. A** Overexpression of hNMNAT2 robustly protects against RGC loss in *Nmnat2*<sup>gtBay/gtE</sup> (25%) retinas maintained for 3 days ex vivo following axotomy, rescuing the RGC sensitization phenotype of these mice (Day 0: *Nmnat2*<sup>gtBay/gtE</sup>, $n = 6$; Day 3: *Nmnat2*<sup>gtBay/gtE</sup>, $n = 6$; *Nmnat2*<sup>gtBay/gtE</sup> + *hNMNAT2*, $n = 5$; scale bar = 20 μm). **B** Overexpression of hNMNAT2 in C57BL/6J mice confers complete protection against RGC loss at 3 days ex vivo ($n = 6$ retinas for all conditions). **C** In the rat ocular hypertension (OHT) model, which recapitulates many features of human glaucoma, significant RGC loss occurs following 14 days of elevated intraocular pressure (OHT) relative to controls (NT, normotensive). Transfection (3 weeks prior to OHT onset) and expression of GFP alone (AAV GFP) does not significantly alter RGC survival, but RGC survival is significantly enhanced with hNMNAT2 expression. This demonstrates that in a complex disease (with many neurodegenerative mechanisms) NMNAT2 gene therapy provides moderate neuroprotection to RGCs (NT $n = 12$ eyes, OHT $n = 9$ eyes, OHT AAV GFP $n = 10$ eyes, OHT hNMNAT2 $n = 8$ eyes). Scale bars = 25 μm in all images. For **A**, **B** and **C**, *$P < 0.05$, **$P < 0.01$, ***$P < 0.001$, NS non-significant ($P > 0.05$); One-way ANOVA with *Tukey's* HSD. For box plots, the centre hinge represents the median with upper and lower hinges representing the first and third quartiles; whiskers represent 1.5 times the interquartile range.

allowed for the preparation of a series of drug-like derivatives, leading to the identification of more potent compounds very rapidly. The removal of metabolically labile hydroxyl groups led to a decrease in activity, but their replacement with fluorine groups on the aromatic ring restored the NAD-boosting activity, potentially increasing the lipophilicity and metabolic stability of the compounds (*e.g.* Compound 17, 20, 50, Fig. 7C, D, Supplementary Fig. 6). A scaffold-hopping strategy led to the identification of three new cores: tetrahydroquinoline (Group 2), benzomorpholine (Group 3), and tetrahydroquinoxaline (Group 5), which showed a higher potency compared to the chromene core and were selected as scaffolds for the design of a new series of analogues. Across these iterations we identified several small molecules with greater potency in increasing NAD than EGCG. We selected ten top candidates (Fig. 8A) for further testing based on their NAD-boosting capacity. All of these top ten compounds retained NAD-salvage pathway specificity as demonstrated by co-incubation with FK866 where the capacity to increase NAD over control was blocked. This was greatest for compounds in Group 4 and 5 (Fig. 8B, Supplementary Data 2). Mitochondrial membrane potential was also assessed in these top compounds (Supp Fig. 7). To test neuron specificity, NAD-boosting effects of the top ten compounds were assessed in dissociated neuron-high (cortex) and neuron-low (liver, muscle, spleen) tissues. Supporting the previous findings with EGCG, the majority of the newly developed compounds retained neuron specificity (Fig. 8C).

In order to elucidate the important structural features for the NAD-boosting activity of these families of compounds, we built a 3D-structure-activity relationship model using the 56 compounds tested. This revealed critical field/steric contributions to the biological NAD-boosting activity, such as a bulky and hydrophobic group in position 2, and a positive field (hydrogen bond donor, positive electrostatic potential) in position 1 (Fig. 8D). This demonstrates that these classes of compounds could be well accommodated in the identified binding pocket on the NMNAT2 surface, making several H-bonds and hydrophobic interactions with the surrounding residues, further supporting a mechanism of action working through NMNAT2. To explore the potential of these compounds to provide neuroprotection we tested 5 of these compounds in the retinal explant model. Two of these compounds (38, 56; Groups 2 and 3) demonstrated a significant, but incomplete, RGC survival compared to untreated controls (Supplementary Fig. 8A, B) and two compounds (54, 55; Group 5) demonstrated a significant, almost complete, protection to RGCs following axotomy (Fig. 8E). Taken together, we demonstrate that NMNAT2 is a druggable target to generate high levels of neuron-specific NAD and that these novel small molecules can provide neuroprotection in the retina.

## Discussion

Our study demonstrates that *NMNAT2* expression is overwhelmingly restricted to RGCs in the retina (although it is present in the majority of neuronal cell types in the body), is highly variable within RGCs and across individuals, and that when expression falls below a critical threshold, RGC degeneration is significantly compounded. This supports NMNAT2 loss (RNA and/or protein) in glaucoma as an important component of neurodegeneration and that individuals on the lower spectrum of NMNAT2 expression may be more susceptible to RGC loss. However, NMNAT2 protein levels in individual RGCs remains to be definitively assessed which will require the development of RGC specific tools to label NMNAT2 or highly specific antibodies to NMNAT2 suitable for protein quantification. Similarly, NMNAT2 expression (RNA and protein) is highly variable in aged postmortem human brains, and further decreased in brains with Alzheimer's disease where lower NMNAT2 is correlated with worse cognition[10]. We demonstrate that reduced *Nmnat2* expression drives susceptibility to axon injury (axotomy) and future experiments could assess this long term in more chronic optic nerve injury models in these mice (*e.g.*

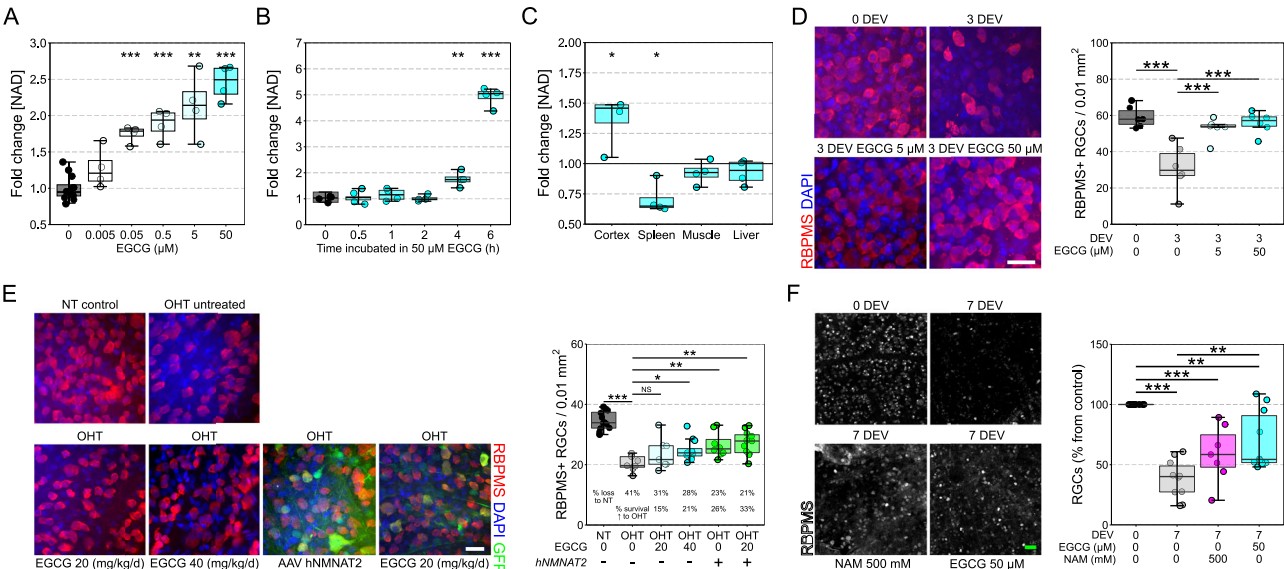

**Fig. 4 | EGCG increases NAD and provides a robust neuroprotection following retinal ganglion cell injury. A** EGCG increases NAD in a dose-dependent manner in dissociated cortical neurons, with 50 nM the lowest dose to give a significant increase in NAD compared to untreated controls (each condition assessed in a sample from the same biological replicate; $n = 4$). **B** EGCG increases NAD in a time dependent manner in dissociated cortical neurons. EGCG was first added to the 6-h samples (and maintained throughout), 2 h later EGCG was added to the 4-h sample, etc. The 0-time sample was incubated for 6 h in media without EGCG. An increased cell viability in the samples treated with EGCG at an earlier time point may also contribute to the 5-fold increase in NAD (each condition assessed in a sample from the same biological replicate; $n = 4$). **C** EGCG significantly increased NAD in neuron-enriched tissue (cortex; $n = 4$ per condition) but not in neuron-low tissues (spleen, muscle, and liver; $n = 4$ per condition) suggesting a specificity towards Nmnat2 over Nmnat1. **D** EGCG dissolved in the culture media robustly protects against RGC death at 3 days ex vivo (3 DEV) following axotomy in comparison to untreated controls ($n = 6$, all conditions) (interventional treatment). **E** In the rat

OHT model, prophylactic oral EGCG provided a modest neuroprotection relative to untreated controls at 40 mg/kg/d ($n = 12$), but not at 20 mg/kg/d ($n = 7$), although this was improved in combination with hNMNAT2 ($n = 11$). **F** In postmortem retinal punches ($n = 10$ donor retinas) maintained ex vivo for 7 days (7 DEV) significant RGC loss occurs which is significantly reduced by EGCG (or nicotinamide, NAM, the precursor for NAD through the NAD-salvage pathway) relative to uncultured controls (0 DEV), supporting the human utility of neuroprotection by EGCG (each condition assessed in a sample from the same biological replicate, $n = 10$). The prolonged postmortem time (24–48 h) results in significant RGC loss, and so in this context EGCG is able to provide interventional neuroprotection to an already degenerating system. Scale bars = 25 μm in D, E and 50 μm in F. *$P < 0.05$, **$P < 0.01$, ***$P < 0.001$, NS non-significant ($P > 0.05$); *Student's* t-test to control for **A**, **B**, and **C**; One-way ANOVA with *Tukey's* HSD for **D**, **E**, and **F**. For box plots, the centre hinge represents the median with upper and lower hinges representing the first and third quartiles; whiskers represent 1.5 times the interquartile range.

optic nerve crush at different ages). Supporting these findings, restoring NMNAT2 expression in depleted retinas via gene therapy removes this susceptibility to RGC degeneration and provides neuroprotection. This neuroprotection is maintained through to a chronic in vivo glaucoma model where the protection is not fully complete likely due to the other OHT-related events (*i.e.* changes in vascular tone, neuroinflammation) which a neuron-specific therapy cannot overcome.

Neuroprotection through similar mechanisms has previously been demonstrated in other neuronal systems through expression of *Wld*[S] (the protein product of which functionally recapitulates the physiological role of NMNAT2[14]) or through modulating the stability and subcellular localization of NMNAT2[23–25]. Similarly, Fang et al.[9] demonstrated that gene therapy delivery of Nmnat2Δex6, a more stable cytosolic form of NMNAT2, is also neuroprotective. In cell culture, Nmnat2Δex6 provides stronger neuroprotection than native Nmnat2 overexpression or Wld[S26]. In this regard, Nmnat2Δex6 behaves functionally similar to the Wld[S] protein[26] which has previously been demonstrated to be robustly neuroprotective in glaucoma[27–29].

A number of NAD precursors have been explored for their potential to increase NAD across different tissues, particularly the CNS where depletion of NAD pools drives, or contributes to, neurodegeneration. Nicotinamide has strong therapeutic potential given its long clinical history, strong neuroprotection in animal models of glaucoma and other neurodegenerative diseases[30], and functional benefits established in short-term clinical trials[31,32]. However, nicotinamide

requires high doses to achieve meaningful NAD increases in the CNS and is not specific to neurons. Whilst nicotinamide riboside (NR) can achieve greater (although more transient) NAD increases than nicotinamide at lower doses[33], its utility in the retina is more limited due to the low expression of NRK (required to convert NR to NMN[6]) which produces only a modest 10–20% NAD increase[34]. Stabilized versions of NMN, while able to raise NAD, are potentially dangerous alternatives in NAD depleted systems (such as in neurodegenerative diseases) given that high ratios of NMN to NAD trigger SARM1 activation and axon degeneration[35–37]. Similarly, SARM1 inhibition through newly developed small molecule inhibitors are strong neuroprotective candidates but are reliant on large pools of NAD[38], which are absent during the neurodegenerative disease cascades. While SARM1 inhibitors will limit NAD consumption, they do not increase new NAD formation and so their long-term use may be limited. Supporting this, *Sarm1*[−/−] does not prevent RGC soma loss following optic nerve crush[39]. In the axon degeneration cascade, NMNAT2 acts up-stream of SARM1[40], and so offers an alternate target, particularly as restoring the capacity of NMNAT2 activity towards normal levels is unlikely to present significant side effect in comparison to removal of a native function (*e.g.* blocking SARM1). In addition, inhibition of SARM1 could have deleterious effects given its protective role in preventing viral spread[41,42]. Small molecules that target NMNAT2 could provide neuronal targeted increases in NAD to boost NAD pools and prevent the initiation of neurodegenerative cascades. This could be relevant in many neurodegenerative diseases in which a depletion of NAD generating capacity or NMNAT2 levels have been demonstrated.

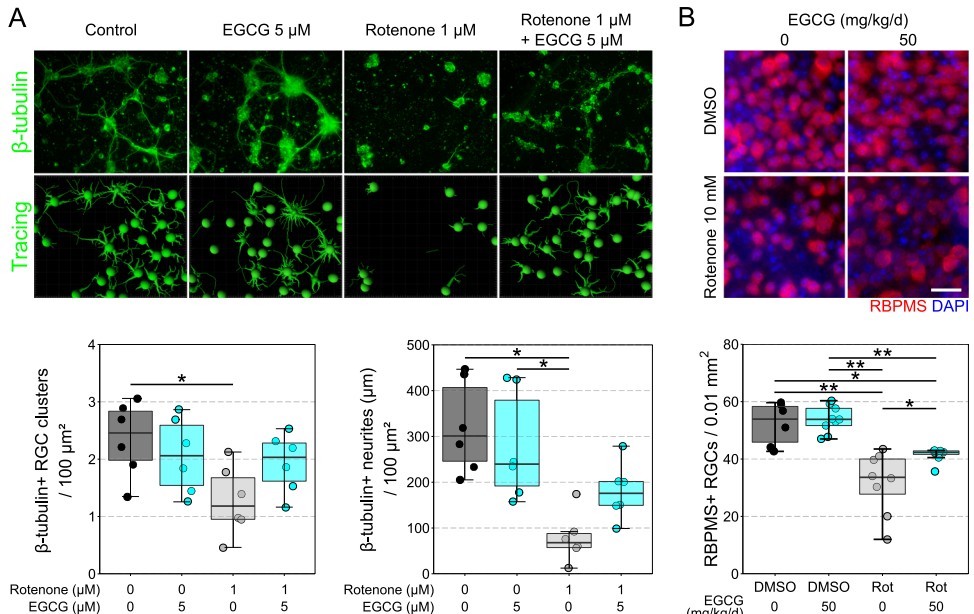

**Fig. 5 | EGCG improves neuronal resilience to rotenone injury. A** Primary retinal neuron cultures were established from P2-3 C57BL/6J mouse pups and grown for 10 days. At day 11, neurons were stressed with rotenone (1 μM) or remained vehicle treated controls (DMSO) in the presence of EGCG (5 μM) or controls (*n* = 6 wells per condition) for 1 day. Rotenone caused a significant decrease in neuron density and total neurite length which was prevented by EGCG treatment. **B** Mice were either treated with EGCG (or untreated) for 1 week prior to receiving an intravitreal injection of 10 mM rotenone or DMSO only (vehicle). RBPMS density was assessed 1 day after intravitreal injection. Rotenone injection resulted in a significant loss of RGCs in untreated mice (~40%) which was significantly mitigated by EGCG treatment (~20% loss). DMSO only *n* = 6 retinas, Rotenone only *n* = 8 retinas, DMSO EGCG *n* = 9 retinas, Rotenone EGCG *n* = 8 retinas. Scale bar = 25 μm for **B**. For **A** and **B**, *\*P* < 0.05, *\*\*P* < 0.01, *\*\*\*P* < 0.001, NS = non-significant (*P* > 0.05); One-way ANOVA with *Tukey's* HSD. For box plots, the centre hinge represents the median with upper and lower hinges representing the first and third quartiles; whiskers represent 1.5 times the interquartile range.

We identified that EGCG can increase NAD and provide neuroprotection against RGC injury across in vitro, ex vivo, and in vivo models (including in ex vivo human tissue). Across multiple methods of raising cellular NAD levels in neural tissue (including via *Nmnat1* gene therapy[3], *Wld*[S] transgene[27], and high dose nicotinamide[3,20]) we have consistently reached a threshold/ceiling of ~200% of control NAD (which is sufficient for strong/complete neuroprotection in the retina and optic nerve). EGCG at μM doses can achieve this level of NAD increase in neuronal tissue to a therapeutic level. Critically, EGCG increased NAD capacity in neuronal, but not non-neuronal, tissues. Supporting that this occurs due to specificity for NMNAT2, EGCG's NAD boosting and neuroprotective effects are reduced or blocked in the presence of the NAMPT inhibitor FK866 or with *Nmnat2* depletion (it is important to note that in these *Nmnat2* depleted systems, ~25% Nmnat2 activity remains. As complete *Nmnat2* KO mice are not viable, further development of conditional tools of Nmnat2 would be of benefit to further confirm these findings). Our in silico data support the existence of a binding site for EGCG on an NMNAT2 homology model. As a polyphenol, EGCG has a number of potential mechanisms of action that could be providing neuroprotection including through modulation of reactive oxygen species, cell signalling, growth factors, autophagy, and apoptotic cascades[43]. Modulation of these could also affect downstream NAD levels[44]. Glaucoma is a complex neurodegenerative disease in which these factors have previously been identified as potential pathological mechanisms[45]. While EGCG did not increase NAD in non-neuronal tissues, this does not preclude other effects derived from these other properties if given as an oral treatment. Considering EGCG's poor bioavailability[22], it would likely have a poor therapeutic index.

Rather than develop a strategy for delivering therapeutic levels of EGCG (*e.g.* via encapsulation or direct targeting to the eye) we instead used EGCG as a tool compound to rationally design novel NMNAT2 targeting small molecules. We confirmed that our top candidates significantly increase NAD capacity in an NMN-dependent manner, supporting specificity to the NAD-salvage pathway, whilst in silico modelling supported the maintenance of binding capability to NMNAT2. Compounds in the later groups of refinement (*e.g.* Group 4 and 5) demonstrated total loss of NAD boosting activity when co-incubated with FK866, more so than for EGCG, further supporting development towards an NMNAT2 specific mechanism of action and loss of other potential mechanisms (*e.g.* reduced effects on mitochondrial membrane potential compared to EGCG). These top compounds were able to provide neuroprotection against RGC degeneration ex vivo, supporting their utility and potential as neuroprotective therapeutics in the future. Next steps could further define the effect of these compounds at a single cell level with specificity to RGCs. This would allow the elucidation as to whether only RGCs require increased NAD generating capacity for neuroprotection, or whether other neurons in the retinal circuitry contribute to this neuroprotection. These experiments could also assess whether subtypes of RGCs are more or less susceptible to low or high NMNAT2 levels.

The small molecules developed here serve as a starting point for further medicinal chemistry to develop NMNAT2-targeted drugs that could be used in vivo, and potentially progress into human clinical trials. The next important steps will be further drug development focusing on ADME and toxicity in rodent and human cells lines, toxicity in vivo, and in vivo pharmacodynamics and pharmacokinetics to establish dose/response ranges which can then be tested in mature, in vivo, animal models of glaucoma with multiple disease metric assessed (*i.e.* visual function, optic nerve and axon morphologies, mitochondrial dynamics). As NMNAT2 has been implicated as a core component of axon degeneration, if properly tested, these compounds have potential for other neurodegenerative diseases.

In conclusion, we demonstrate that *NMNAT2* is highly variable within neuronal populations of the visual system, and that low levels of

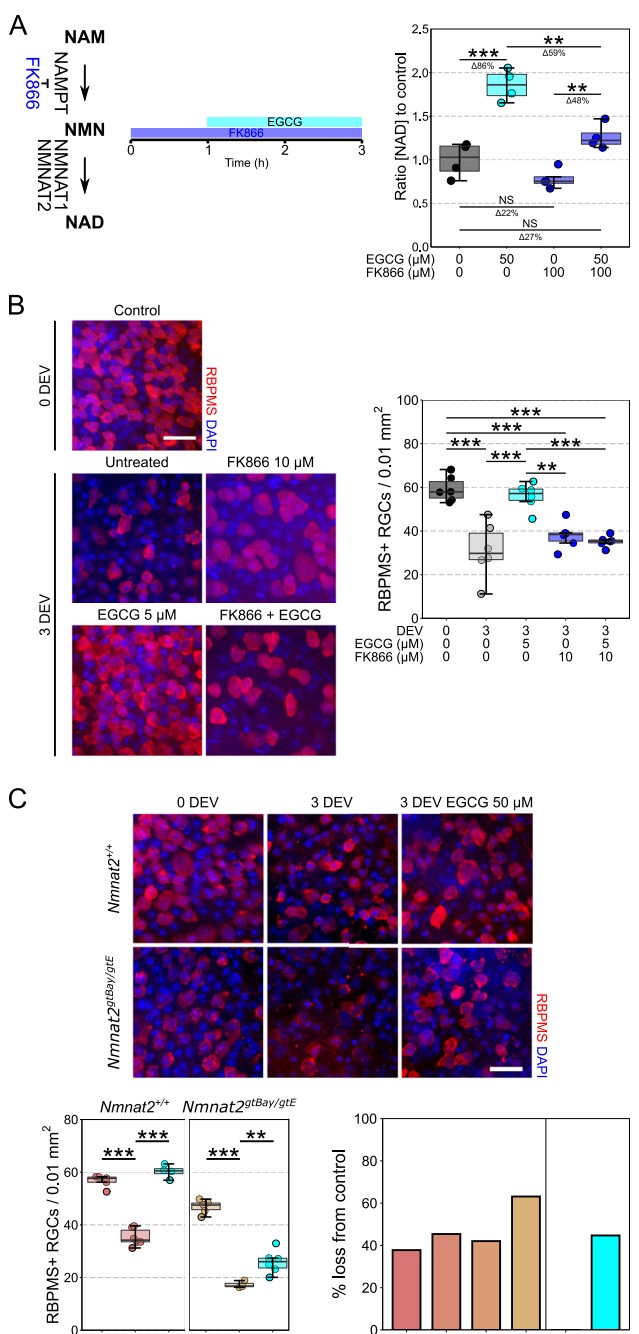

**Fig. 6 | EGCG increases NAD and provides neuroprotection through an NMN and NMNAT2-dependent mechanism. A** Addition of FK866 (an NAMPT inhibitor which reduces the levels of NMN available to NMNAT2) significantly decreases the capacity of EGCG to produce additional NAD ($n = 4$ cortex for all conditions). **B** In the retinal axotomy model addition of FK866 does not significantly alter RGC survival compared to untreated controls. However, the neuroprotective effect of EGCG was completely abolished in the presence of FK866, suggesting that in the context of a RGC injury, EGCG's neuroprotective effect is derived through an NMN-dependent mechanism ($n = 6$, all conditions). **C** Supporting this, in *Nmnat2*[+/+] mice (100% *Nmnat2*) EGCG provides complete neuroprotection at 3 DEV ($n = 4$), but in *Nmnat2*[gtBay/gtE] mice (25% *Nmnat2*, $n = 6$), the neuroprotective effects of EGCG are significantly diminished (44% RGC loss, which is comparable to untreated *Nmnat2*[+/+], *Nmnat2*[+/gtBay], and *Nmnat2*[+/gtE] mice). This suggests that the neuroprotective effects of EGCG work through an NMNAT2-dependent mechanism. Scale bars = 25 μm in B and C. For **A**, **B**, and **C**, *$P < 0.05$, **$P < 0.01$, ***$P < 0.001$, NS = non-significant ($P > 0.05$); One-way ANOVA with *Tukey's* HSD. For box plots, the centre hinge represents the median with upper and lower hinges representing the first and third quartiles; whiskers represent 1.5 times the interquartile range.

NMNAT2 render RGCs susceptible to neurodegenerative insults. Increasing NMNAT2 via gene therapy or increasing NMNAT2's NAD output pharmacologically robustly protects from neurodegenerative insults. Using EGCG as a tool compound we designed, synthesized, and tested novel small molecules that drive NAD production with specificity to the NAD-salvage pathway and provide neuroprotection. NMNAT2 targeting small molecules are ideal opportunities for clinical translation in neurodegenerative diseases and aging.

## Methods

### Ethics statements
Individual study protocols were approved by Stockholm's Committee for Ethical Animal Research (10389-2018) or the UK Home Office (PPL P98A03BF9). Access to human eye tissues and patient samples is fully covered under "Studier av neuronal metabolism, biomarkörer och neuroprotektion vid glaukom" 2021-01036 (additions to base application 2020-01525; Pete Williams).

### Animal strain and husbandry
All breeding and experimental procedures were undertaken in accordance with the Association for Research for Vision and Ophthalmology Statement for the Use of Animals in Ophthalmic and Research. Animals were housed and fed in a 12 h light/12 h dark cycle with food and water available ad libitum. Male Brown Norway rats (*Rattus norvegicus*) aged 16–20 weeks were purchased from SCANBUR and housed for 1 week before beginning experiments. Male C57BL/6J mice (B6, SCANBUR) were purchased at 10–12 weeks old and housed for 1–4 week before beginning experiments. *Nmnat2* gene-trap allele mouse lines *Nmnat2*[+/gtBay] and *Nmnat2*[+/gtE] (as previously described[15]) were kindly provided by Michael P. Coleman. *Nmnat2*[+/gtBay] and *Nmnat2*[+/gtE] were subsequently bred in house (Stockholm) to produce *Nmnat2*[+/+], *Nmnat2*[+/gtBay], *Nmnat2*[+/gtE], and *Nmnat2*[gtBay/gtE] genotypes. *Nmnat2* mice (mixed sex) were used at 3, 6, and 12 months old. Tissue from mice aged to 22-month-old were from the original Cambridge colony housed in 12 light/dark with food and water available ad libitum. Mice were euthanized and eyes were shipped to Stockholm for analysis.

### *NMNAT2* expression variability
To determine the variability of expression of *NMNAT2* among genetically diverse individuals we used publicly available data from Gene-Network (www.genenetwork.org[46]). For mice, BXD family populations were assessed for *Nmnat2* expression across available visual system relevant datasets. We queried *Nmnat2* mRNA expression in the eye ($n = 157$; UTHSC BXD All Ages Eye RNA-Seq (Nov20) TPM Log2), in the retina in young animals <100 days old ($n = 55$, DoD CDMRP Retina Affy MoGene 2.0 ST (May15) RMA Gene and Exon Level), and in the midbrain ($n = 37$) which contains the superior colliculus, where ~80% of RGCs project to in the mouse (VU BXD Midbrain Agilent SurePrint G3 Mouse GE (May12) Quantile). We determined whether *Nmnat2* expression was related to RGC variability by performing a *Spearman*'s rank correlation test comparing the average *Nmnat2* expression in the retina from GeneNetwork (DoD CDMRP Retina Affy MoGene 2.0 ST (May15) RMA Gene and Exon Level) to the average RGC density of each strain from Williams et al.[11]. From the same dataset, the variance in expression of *Nmnat2* was compared to RGC markers *Pou4f1*, *Rbpms*, *Thy1*, and *Tubb3* by performing Paired *Pitman-Morgan* tests using the *var.test* function in the *PairedData* package for R. We also queried how *Nmnat2* expression in the eye was changed by age (UTHSC Individual BXD Adult and Aged Eye RNA-Seq (Dec20) TPM Log2) where mice were grouped by age to either <200 days ($n = 157$) or >350 days ($n = 187$). For human *NMNAT2* expression in the retina, we queried retinal *NMNAT2* mRNA across a sample of 50 individuals (TIGEM Human Retina RNA-Seq (Sep16) RPKM log2) and whole retina bulk RNA-sequencing data from The Genotype-Tissue Expression (GTEx) Project through The Human Protein Atlas [accessed 11/22/2022]. *NMNAT2* expression

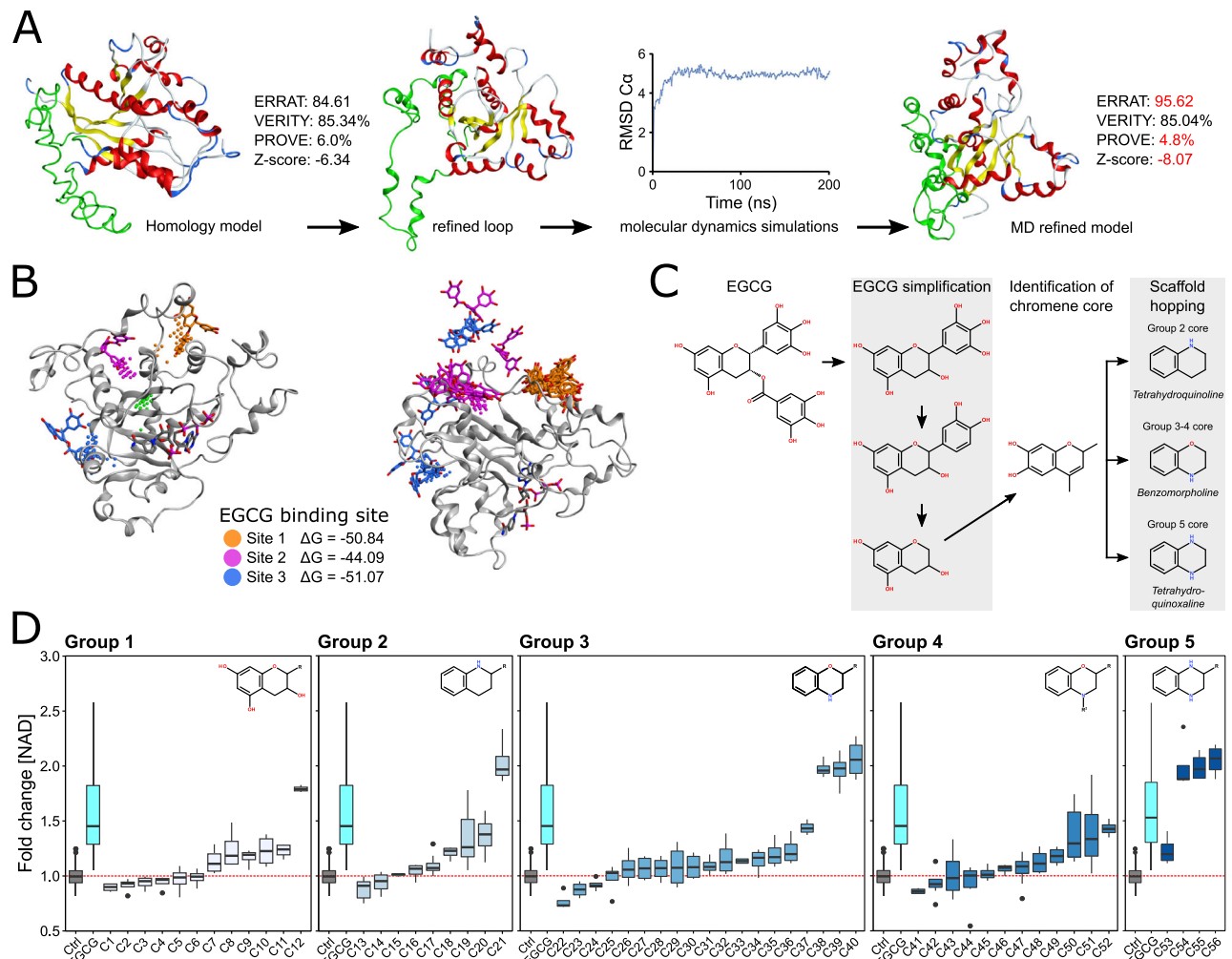

**Fig. 7 | EGCG provides a basis for generating novel NAD-producing compounds.**
**A** An NMNAT2 homology model using NMNAT1 and NMNAT3 was further refined using loop-modelling software (DaReUS-Loop) to refine a low homology domain in the central region of the protein (green). Molecular dynamic simulations demonstrated that the protein reached a stable conformation after 100 ns. The final generated protein conformations demonstrated greater reliability (ERRAT, VERIFY 3D, PROVE, and Z-score). **B** Three potential druggable binding pockets (independent of the NMN catalytic pocket) were identified. The docking pose of EGCG was well accommodated in these three different druggable pockets with corresponding ΔG scores < −40 kcal/mol (*left*). Only one ligand-protein complex (Site 1; orange) could maintain a stable conformation over a 500 ns molecular dynamic simulation (*right*), demonstrating in silico evidence that EGCG can directly bind to NMNAT2.

**C** Given its poor drug-like properties, EGCG was used as a tool compound to identify novel NAD-producing compounds. The EGCG structure was truncated in series to identify a biologically active core which was then used in a scaffold-hopping strategy. **D** An iterative synthesis and NAD-testing (luminometry of dissociated cortex neurons) pipeline was established to identify and test NAD-producing compounds. A number of compounds with greater efficacy than EGCG at producing NAD were identified ($n = 4$/condition; statistical testing in Supplementary Data 1). Outliers denoted by black circles. *$P < 0.05$, **$P < 0.01$, ***$P < 0.001$, NS non-significant ($P > 0.05$). For box plots, the centre hinge represents the median with upper and lower hinges representing the first and third quartiles; whiskers represent 1.5 times the interquartile range.

variability within individual RGCs was assessed using single-cell RNA-sequencing data from Gautam et al.[47], (GEO: GSE147979) and single-nucleus RNA-sequencing data from Orozco et al.[48], (GEO: GSE135133). The datasets was processed using Seurat V3.2. QC was performed to remove doublets and empty droplets by filtering cells with UMI 1000–6000 and detected genes with 500–5000. Clustering and cell annotation ID provided by the authors were used to identify the major cell types in the retina. To compare *NMNAT2* expression across individual RGCs, the *NormalizeData()* function was used to generate normalized and log-transformed single-cell expression with a scale factor of 10,000. To compare *NMNAT2* expression across all retinal cell types data were scaled using the *ScaleData()* function following identification of highly variable features using the *FindVariableFeatures()* function (vst method; 2000 features). *Nmnat2* expression across glaucoma progression was determined from publicly available microarray gene-expression data from whole optic nerve head (ONH) and whole retina

in 10.5-month-old DBA/2J (D2) mice[12,13] (GSE26299). We explored *Nmant2* expression across the disease clusters identified by Howell et al.[12] which are based on the degree of genetic change (and correspond to the degree of neurodegeneration in histological optic nerve analysis). In the ONH, these grouping are: Group 1 = no detectable glaucoma, limited genetic change, ($n = 8$); Group 2 = no detectable glaucoma, significant genetic change, ($n = 8$); Group 3 = no or early glaucoma, significant genetic change, ($n = 6$); Group 4 = moderate glaucomatous degeneration, significant genetic change, ($n = 4$); Group 5 = severe glaucomatous degeneration, significant genetic change, ($n = 4$). Expression is compared to $n = 5$ D2-Gpnmb + . In the retina, there are 4 groups: Group 1 = no detectable glaucoma, limited genetic change, ($n = 8$); Group 2 = no detectable glaucoma, moderate genetic change, ($n = 9$); Group 3 = moderate glaucomatous degeneration, significant genetic change, ($n = 3$); Group 4 = severe glaucomatous degeneration, significant genetic change ($n = 10$). Expression is

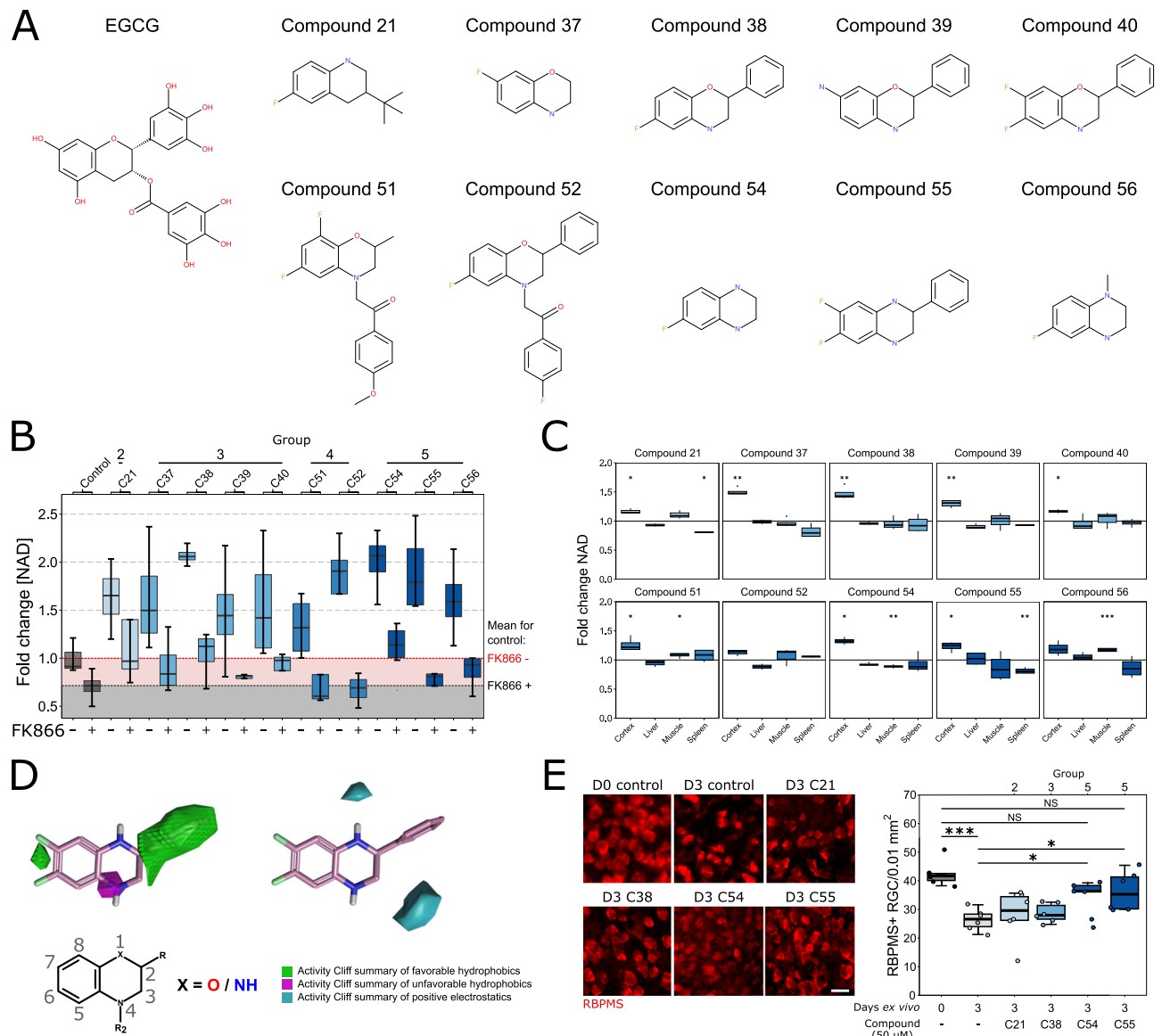

**Fig. 8 | Generation of NAD-salvage pathway specific compounds that drive NAD production and provide neuroprotection. A** We selected 10 compounds for further testing based on their NAD-producing capacity (structures shown in comparison to EGCG). **B** For 9 of the 10 top compounds, FK866 suppressed the NAD-boosting effect, demonstrating that these drugs retained a mechanism of action through the NAD-salvage pathway ($n = 4$ cortex for all conditions; fold change in NAD comparable to normal controls, *denoted by red space*; fold change comparable to FK866 treated normal controls, *denoted by black space;* statistical testing in Supplementary Data 2). **C** Compounds were tested for NAD modifying capacity in dissociated cortex, liver, muscle, and spleen ($n = 4$ for all conditions). An NAD-boosting effect in neuron-low tissue was only demonstrated for 2 compounds (51, 56) with both increasing NAD in muscle by <1.2 fold relative to untreated controls. A reduction in NAD relative to control was identified for 3 compounds in neuron-low tissue, with compound 21 and 55 reducing NAD in spleen to -0.8 fold, and

compound 54 reducing NAD in muscle to -0.9 fold. Results are normalized to untreated controls of matched tissue type. **D** The compounds were used to generate a structure-activity relationship model which identified favourable (green, at C2) and unfavourable (purple) hydrophobic regions which affect the activity, and negative electrostatics regions (blue, at C1) which are crucial for NAD-boosting activity. **E** Five of the top ten compounds were tested for neuroprotective capacity in a retinal explant model. Compounds 54 and 55 (from group 5) demonstrated a significant protection of RGCs, demonstrating the potential of these compounds to provide neuroprotection ($n = 6$ retina/condition). Scale bar = 20 μm. For **B**, **C**, and **E**, *$P < 0.05$, **$P < 0.01$, ***$P < 0.001$, NS non-significant ($P > 0.05$), *Student's* t-test to control. For box plots, the centre hinge represents the median with upper and lower hinges representing the first and third quartiles; whiskers represent 1.5 times the interquartile range.

compared to $n = 8$ D2-Gpnmb + . *Nmnat2* expression in RGCs early in glaucoma (9-month-old D2) was explored using publicly available data from Williams et al.[3]; all mice used for these experiments were confirmed to have no detectable neurodegeneration in the optic nerve ($n = 9$ D2-Gpnmb+ and $n = 26$ D2). Grouping represents degree of genetic change (1 = least, 4 = most): Group 1 ($n = 9$ age-matched D2-Gpnmb+ and $n = 6$ D2s), Group 2 ($n = 6$ D2s), Group 3 ($n = 10$ D2s), Group 4 ($n = 4$ D2s).

## Human donor retina
For ex vivo human retina, de-identified eyes were acquired from the St. Erik eye hospital corneal eye bank as waste tissue following donation for corneal transplant. Only eyes free from known ocular or metabolic disease were selected. Upon receipt, retinas were dissected free and the vitreous was removed. Donor tissue details are recorded in Supplementary Table 1. Retinal punches were taken (Sterile Disposable Biopsy Punch, 2 mm diameter, MILTEX) in an arc 3 mm nasally from the

optic nerve head. Punches were used for a retinal explant model (described below).

## Retinal explant model

Retinal explants were performed as previously described for mice[20] and human retinas[49]. Flat mounted retinas were maintained in culture (37 °C, 5% $CO_2$) fed by Neurobasal-A media supplemented with 2 mM L-glutamate (GlutaMAX, Gibco) 2% B27, 1% N2, and 1% penicillin/streptomycin (all Gibco) in 6-well (mouse) and 24-well culture plates (human) with corresponding cell culture inserts (Millicell 0.4 μm pore; Merck). For mice, animals were euthanized by cervical dislocation, retinas dissected free in cold HBSS and flat mounted on cell culture inserts ganglion cell layer up. Retinas were removed from culture and fixed in 3.7% PFA at 3 days ex vivo for cell counts or homogenized at 12 h ex vivo for qPCR (see below for details). Control eyes were processed immediately following enucleation. For cell counts, retinas were dissected following 1 h of fixation. For human retina, 2 punches were immediately fixed in 3.7% PFA (control) following dissection, and 6 were explanted on to individual cell culture inserts ganglion cell layer up (2 untreated, 2 NAM treated, 2 EGCG treated). Punches were maintained in culture ex vivo as described above for 7 days before fixation in 3.7% PFA. Media was changed at day 2 for the mouse, and on alternate days for the human. For drug treatments, drugs were dissolved in the culture media to a concentration of: EGCG (5, 50 μM for mouse; 50 μM for human; Merck), NAM (500 mM for human; PanReac AppliChem), whole green tea polyphenol extract (0.1 μg, 1 μg for mouse; Abcam), FK866 (10 μM for mouse; Sigma). Novel compounds were diluted to 0.5 μM, 5 μM, and 50 μM in DMSO.

## Rat bead model and drug treatments

Prior to induction of ocular hypertension rats were habituated to intraocular pressure (IOP) measurement by rebound tonometry (Tonolab, Icare). Baseline IOP was recorded the morning before surgery (day 0). For drug treatments, rats received drug dissolved in drinking water 1 week prior to OHT induction, with treatment continuing for the remainder of the experiment (14 days). Drug doses were calculated based on an average daily intake of 15 ml of water per rat. The effective EGCG dose was ~20 or 40 mg/kg/d. Whole green tea polyphenol extract was given to achieve a dose of ~80 mg/kg/d. The effective whole green tea polyphenol dose was ~80 mg/kg/d. Ocular hypertension was induced using a paramagnetic bead model as previously described[50]. Rats were anaesthetized with an intraperitoneal injection of ketamine (37.5 mg/kg) and medetomidine hydrochloride (1.25 mg/kg). Microbeads (Dynabead Epoxy M-450, Thermo Fisher) were prepared in Hank's balanced salt solution (HBSS -$CaCl_2$ -$MgCl_2$ -phenol red, Gibco) and 6–8 μl of bead solution was injected into the anterior chamber. Beads were distributed using a magnet to block the iridocorneal angle. Rats received either bilateral injections (OHT) or remained bilateral unoperated (naïve), normotensive controls (NT). IOP was measured at day 3, 7, 9, 11, and 14 post induction of OHT. IOP recordings were always performed between 9 and 10 am to avoid the effects of the circadian rhythm on IOP. Rats were taken to either 7 days or 14 days as an end-point.

## Gene therapy

Gene therapy was delivered through intravitreal injection of AAV2-CMV-hNMNAT2-CMV-eGFP or AAV2-CMV-eGFP as a control. Mice or rats were anaesthetized by intraperitoneal injection of ketamine and medetomidine hydrochloride as above. Bilateral intravitreal injections were performed using a 33 G tri-beveled needle on a 10 μl glass syringe (WPI). AAV diluted to $2.2 \times 10^{11}$ GC/ml or AAV2-CMV-hNMNAT2-CMV-eGFP $7.8 \times 10^{11}$ GC/ml AAV2-CMV-eGFP in HBSS was injected into the vitreous (1 μl for mice, 3 μl for rats) and the needle maintained in place for 30 s for distribution. Three weeks was allowed to achieve sufficient transduction and expression before animals were used for either the retinal explant model or the rat bead model.

## Intravitreal rotenone model

Rotenone was delivered in vivo to the retina through intravitreal injection[20]. B6 mice were pretreated with EGCG dissolved in drinking water to give a dose of ~50 mg/kg/d or untreated. Intravitreal injections of either 1 μl of a 10 mM rotenone (MP Biochemicals) solution dissolved in DMSO (PanReac AppliChem) or DMSO only (vehicle only control) were performed in EGCG and untreated mice. Mice were euthanized 1 day following injection and retinas were processed for immunofluorescent labelling of RGCs.

## Primary retinal neuron culture

For primary retinal neuron cultures P2-3 B6 mice were euthanized by decapitation and retinas were dissected. To dissociate cells, 6 retinas were pooled in 1 ml of dispase (500 U, Corning) and maintained at 37 °C for 45 min in a heating block (Thermomixer C, Eppendorf) set at 350 rpm. Cells were pelleted and resuspended in culture media for seeding onto Poly-D-Lysine coverslips (Corning) in a 24-well plate. To encourage the selective growth and maintenance of neurons, cells were cultured in Neurobasal-A media supplemented with 2 mM L-glutamate, 2% B27, 1% penicillin/streptomycin, and 50 ng/ml BDNF. Cells were cultured for 10 days with a media change every 2 days. At day 10, neurons were stressed with rotenone (1 μM in DMSO) or DMSO only and the media was supplemented with EGCG (5 μM) or remained untreated. At day 11, neurons were fixed with ice cold methanol for 20 min and processed for immunofluorescent labelling in the wells. Neurons were permeabilized in 0.5% TritonX for 5 min, blocked in 1% BSA for 30 min, incubated with anti-βIII-tubulin (NB100-1612, Novus-Biological). Neurons were washed 3× in PBS for 5 min, incubated with secondary antibody for 1 h and washed 3× in PBS for 5 min. Coverslips were removed and inverted on to glass slides with Fluoromount-G mounting medium. Six images per coverslip were acquired on a Zeiss Axioskop 2 plus epifluorescence microscope (Karl Zeiss) at 40×. Using Imaris (Bitplane, version 9.3.1) neuron morphology was reconstructed with the filament function to calculate the number of neuron clusters and total neurite length. Values per sample (coverslip) were taken as the mean of six images.

## qPCR

Rat optic nerves were collected following euthanasia at 7 days post-OHT induction (and NT controls) using pentobarbital (75 mg/kg) followed by cervical dislocation. The brain was removed and the optic nerves cut at the chiasm. Optic nerves were cut to 4 mm from the end proximal to the eye before flash freezing on dry ice. Before use, optic nerves were thawed on ice and homogenized and sonicated in 350 μl DNAse free water for 20 s, 30,000 $min^{-1}$ (VDI 12, VWR). Half of the sample was combined with 150 μl of 2× buffer RLT (Qiagen) with 2% β-mercaptoethanol (Fisher Scientific). Flat mounted retina were lifted from culture inserts by gentle agitation with HBSS and homogenized into 400 μl buffer RLT (Qiagen) with 1% β-mercaptoethanol (Fisher Scientific) using a QIAshredder kit (Qiagen) according to the manufacturer's instructions. For all tissue, RNA was extracted using RNeasy Mini Kits (Qiagen) according to the manufacturer's instructions. RNA was extracted into nuclease-free water, and RNA concentration was measured in a 1 μl sample diluted 1:200 in nuclease-free water in a spectrophotometer (BioPhotometer, Eppendorf). cDNA was synthesized using 1 μg of input RNA with an iScript™ cDNA Synthesis Kit and MyIQ thermocycler (both Bio-Rad) and stored at −20 °C overnight. RT-qPCR was performed using 1 μg of input cDNA, 7.5 μl of SsoAdvanced Universal SYBR Green Supermix and 1 μl of the following DNA templates (Prime PCR Assay, Bio-Rad): *Nmnat2* (qMmuCID0005266) and *Rps18* (housekeeping; qMmuCED0045436) in the mouse; *Nmnat2* (qRnoCID0001493), and *GAPDH and TBP* (housekeeping; qRnoCID0057018, qRnoCID0057007) in the rat (all templates were species specific—*Mus musculus*, *Rattus norvegicus*, respectively; Bio-Rad). A MyIQ

thermocycler was used with a 3 min activation and denaturation step at 95 °C, followed by an amplification stage comprising 50 cycles of a 15 s denaturation at 95 °C and 1 min annealing and plate read at 60 °C. Analysis was performed according to the ΔΔCT method.

## Immunofluorescent labelling of RGCs

Following fixation, retinal flat mounts or punches were mounted on slides (histobond) and isolated using a hydrophobic barrier pen (VWR). Retinas were permeabilized with 0.5% Triton X-100 (VWR) in PBS for 30 min, blocked in 2% bovine serum albumin (BSA; Fisher Scientific) in PBS for 30 min, and primary antibody were applied and maintained overnight at 4 °C. Following five repeated washes of 5 min with PBS, secondary antibodies were applied and slides were maintained at room temperature for 2 h. Tissue was washed as before and DAPI nuclear stain (1 μg/ml in PBS) was applied for 10 min. Tissue was washed once in PBS before being mounted using Fluoromount-G and glass coverslips were applied. Slides were sealed with nail varnish. The immunofluorescent labelling of the retinas treated with C21, C38, C54, C55 and C56 in the retinal explant model followed this protocol but with the following variations: retinas were permeabilized with 0.1% Triton X-100 in PBS for 60 min and blocked in 2% BSA in HBSS for 60 min. Primary antibody was applied and maintained over three nights at 4°C, following five repeated washes for 10 min, secondary antibody was applied and maintained for 4 h at RT. Retinas were washed 5 times for 10 min before staining with 5 μg/mL nuclear Hoechst 33342 nuclear stain (Thermofisher Scientific) diluted in PBS. Primary antibodies used were: anti-RBPMS (NovusBiological, cat #NBP2-20112, lot # 42858; 1:500) with or without anti-GFP (Abcam, cat # ab13970, lot #, GR3190550-8 and 1018753-4; 1:500) for retinas which had received gene therapy. RGC density was assessed by counting RBPMS+ cells in the GCL. Images were acquired on either a Zeiss Axioskop 2 plus epifluorescence microscope or Zeiss LSM800-Airy (both Karl Zeiss). Six images per retina were taken equidistant at 0, 2, 4, 6, 8, and 10 o'clock about a superior to inferior line through the optic nerve head ( -1000 μm eccentricity). Images were cropped to $100 \times 100$ μm and RBPMS+ cells were counted using the cell counter plugin for Fiji; counts were averaged across the 6 images. Cells were counted by a minimum of two observers blinded to treatment.

## Luminescence-based NAD assay

B6 mice were euthanized by cervical dislocation, and the whole cortex was removed and separated by hemisphere. For experiments where other tissues were assessed, whole spleen, the left liver lobe, and hamstring muscles were dissected and collected. Tissue was stored on ice in HBSS until the next step. Cortical hemispheres (maintained separately) were submerged in 800 μl dispase (5000 Caseinolytic units, 354235, Corning). Spleen was cut into two pieces and the pieces were put separately in 800 μl dispase. The left liver lobe was cut into two pieces and the pieces were separately added to 800 μl dispase, and each hindlimb muscle was a separate sample added to 800 μl dispase. Tissues were finely chopped in dispase and were put on a heating block (ThermoMixer® C, Eppendorf) at 37 °C, 350 rpm for 30 min before dissociation by gentle trituration. Cell concentration of the cell suspension was determined with a hemocytometer (C-Chip, NanoEntek). Cortical, spleen, liver, and muscle samples were diluted to a concentration of 2 million cells/ml with HBSS (in which 100,000 cells were used per assay well). (Due to the low yield ( > 50,000 RGCs/retina), NAD levels in purified RGCs were not assessed in these studies.) Samples were incubated with compounds of interest for 2 h at 37 °C and 5% $CO_2$ (solutions were diluted with HBSS from 50 mM stock solution in DMSO (A3672-0100, PanReac AppliChem)). After incubation, samples were homogenized for 20 s with a handheld homogeniser (30.000 min⁻¹, VDI 12, VWR). Compounds which displayed activity at 5 μM but not at 50 μM were

re-tested in an updated protocol where, following treatment, samples were spun down at 7700 rpm (4000 g) for 5 min, and the supernatant was removed and exchanged with HBSS prior to homogenization to remove drugs that may interfere with the luminometric signal. To detect NAD, detection, the NAD reagent was prepared according to the manufacturer's protocol (NAD/NADH-Glo™, Promega). 50 μl of the sample was combined with 50 μl of reagent in a 96-well plate (Nunc™ F96 MicroWell™ White Polystyrene plate, ThermoFisher Scientific) and luminescence was recorded with a plate reader (infinite® 200, Tecan) 1 h from initial mixing. Fold changes in samples were compared to the luminescence signal in control samples. To determine the effect of EGCG on NAD over time the experiment followed the same procedure as above, except for incubation time. After dissociation and diluting cell samples, EGCG was first added to 6-h samples, and 2 h later EGCG was added to the 4-h sample, etc. The 0-h samples were incubated for 6 h in HBSS without EGCG for 6 h. All samples were incubated at 37 °C and 5% $CO_2$ throughout the time. After incubation samples were processed in the same way as previously stated to assess NAD levels.

## Measuring mitochondrial membrane potential with JC-1

Mitochondrial membrane potential change of mouse cortical cells after treatment with EGCG and compound using JC-1 as previously described[51]. Four hemispheres from whole cortexes were collected and dissociated as described above. Samples were diluted to 1 million cells/ml in HBSS and incubated with EGCG, C21, C37, C38, C39, C40, C51, C52, C54, C55, C56 at 5 μM and 50 μM for 1.5 h (37 °C, 5% $CO_2$). JC-1 (200 μM in DMSO) was added to a final concentration of 2 μM to the samples and the samples were further incubated for 30 min (37 °C, 5% $CO_2$). Cells were spun down at 7700 rpm (4000 g, MiniSpin®, Eppendorf) and resuspended in 1 mL HBSS. 50 μl of each sample was loaded on a 96-well plate (Nunc™ F96 MicroWell™ White Polystyrene plate, Thermo Fisher Scientific) and the fluorescence was measured using excitation/emission at 485/535 nm and 535/590 nm (infinite® 200, Tecan). Mitochondrial membrane potential difference (ΔΨ) was calculated from the ratio between the two wavelength measurements.

## EGCG pH stability

EGCG (pharmaceutical secondary standard, Sigma Aldrich), was diluted in DMSO to 100 mM and diluted to 1 mM in HBSS (Gibco) adjusted to pH 1.8, 3, 4, 5,6 and non-adjusted HBSS (pH 7.6) and kept at 37 °C. The absorbance was measured over 190–850 nm with a micro-UV/Vis spectrophotometer (NanoDrop™ OneC, ThermoFisher Scientific) at time point 0, 0.5, 1, 2 and 24 h.

## Molecular dynamic simulation

The 3D protein structure was modelled using the I-TASSER[52] server and the sequence of the human NMNAT2 (UniProt code Q9BZQ4). NMNAT1 and NMNAT3 were used as templates to guide the NMNAT2 protein design as has previously been used[53]. The low homology domain in the central region of the protein (111–190) was refined using DaReUS-Loop[54] and successively the entire protein was subject to molecular dynamic (MD) simulation using Desmond packaged (Maestro, Schrödinger 2022-2, New York, NY, USA), employing OPLS4 force field in the explicit solvent and the TIP4D water model. A cubic water box was used for the solvation of the system, ensuring a buffer distance of approximately 12 Å between each box side and the complex atoms. The systems were neutralised by adding 6 sodium counter ions. The system was then minimized and pre-equilibrated using the default NPT relax protocol in Desmond. A 1 μs MD simulation in NPT ensembles at constant temperature (300 K) and pressure (1 atm). Data were collected every 100 ps. Hierarchical clustering based on the structural root-mean-squared distance (RMSD) of Cα was used to group the different protein conformations. The three most populated

clusters were used to select three representative structures for subsequent identification of binding sites. Molecular Operating Environment (MOE) 2022 (2022.02, Chemical Computing Group ULC, Montreal, QC, Canada) was used to visualize the protein structures.

## NMNAT2 structure validation

ProsaWeb[55] and SAVES 6.0 (https://saves.mbi.ucla.edu/) were as used to evaluate the models generated through the modelling process. Each test evaluates a different characteristic of the protein models based on current information known about protein structures. These scores were used to compare and rank the protein models.

## Pocket identification, docking and validation

Each representative structure was processed using Schrödinger Site-Map (Schrödinger Release 2022-2: SiteMap, Schrödinger) module to identify potential protein binding pockets. Sitemap generated different descriptors, which define the size, volume, degree of enclosure, hydrophobicity and hydrophilicity of each pocket. All these descriptors contribute to generating a Dscore and Sitecore, which assess the druggability of the pockets. A consensus of the different identified pockets among the three different representative structures was used to select the best 3 binding pockets. For each identified pocket a 12 Å docking grid was prepared for subsequent docking studies. The ECGC molecule was first prepared considering the ionization states at pH $7 \pm 2$ and then subject to docking studies using Schrödinger Glide SP modules precision keeping the default parameters and setting (Schrödinger Release 2022-2: Glide, Schrödinger). Molecular mechanics generalized Born surface area (MMGBSA) was used to re-score the three output docking poses of each compound. Only the best-scored pose for each docking was used for successively MD simulation to assess the ligand-protein complex stability. The MD simulation was carried out using a similar approach and was performed using the same protocol in Desmond described above. The RMSD, pocket occupancies and protein-ligand interactions analyses were performed using the Simulation Interaction Diagram of Desmond. The ΔG binding values of the protein–ligands complex was calculated using the MM/GBSA each 2.5 ns during the entire MD simulations.

## Statistical analysis

All statistical analyses were performed in R. Data were tested for normality with a *Shapiro−Wilk* test. Normally distributed data were compared by *Student's t*-test (one sided) or ANOVA (with *Tukey's* HSD). Non-normally distributed data analysed by a *Kruskal Wallis* test followed *Dunn's* tests with *Benjamini and Hochberg* correction. Unless otherwise stated, *$P < 0.05$, **$P < 0.01$, ***$P < 0.001$, NS = non-significant ($P > 0.05$). For box plots, the centre hinge represents the median with upper and lower hinges representing the first and third quartiles; whiskers represent 1.5 times the interquartile range.

## Reporting summary

Further information on research design is available in the Nature Portfolio Reporting Summary linked to this article.

## Data availability

The data generated in this study are provided in the Supplementary Information/Source Data file. NMNAT2 expression data are publicly available from GeneNetwork (www.genenetwork.org) under gene network accession codes GN1027, GN709, GN381, GN802. Whole retina bulk RNA-sequencing data is available from The Genotype-Tissue Expression (GTEx) Project through The Human Protein Atlas [accessed 11/22/2022]. RGC single cell and single nucleus RNA-seq data are available through the Gene Expression Omnibus accession numbers GSE147979 and GSE135133. NMNAT2 expression from DBA/2 J mice are available through the Gene Expression Omnibus accession numbers GSE26299 and GSE90654. For protein modelling, the

sequence of human NMNAT2 used is accessible with UniProt code Q9BZQ4. NMNAT1 and NMNAT3 structure used are available through the RCSB Protein Data Bank, PDB code 1KQN17 for NMNAT1 and 1NUU for NMNAT3. Source data are provided with this paper.

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

## Acknowledgements

The Authors would like to thank Amelie Botling Taube for assistance with writing and submitting ethics for donor human tissue use, as well as Flavia Plastino, Helder André, and Virpi Luoma (and the St. Erik Eye Hospital tissue lab/cornea transplant service) for assistance in acquiring donor human retina for explant experiments. We thank the staff at the Division of Eye and Vision's animal facility for their assistance in animal breeding and husbandry. We thank Rob Williams for his suggestions and assistance regarding GeneNetwork. J.R.T is supported by Ögonfonden, KI Foundation Grants for Eye Research, Loo och Hans Ostermans stiftelse, Stiftelsen Lars Hiertas Minne, and St. Erik Eye Hospital philanthropic donations. A.L is supported by Wellcome Trust. J.G is supported by BBSRC. G.J is supported by the Knut and Alice Wallenberg Foundation and Region Västerbotten. R.C.B.W is supported by the National Health and Medical Research Council (Ideas grant) and Centre for Eye Research Australia Foundation. M.P.C is supported by the John and Lucille van Geest Foundation. P.A.W is supported by Karolinska Institutet in the form of a Board of Research Faculty Funded Career Position, St. Erik Eye Hospital philanthropic donations, Vetenskapsrådet (2018-02124

and 2022-00799), StratNeuro StartUp grant, Ögonfonden, Stiftelsen Lars Hiertas Minne, Stiftelsen Kronprinsessan Margaretas Arbetsnämnd för Synskadade, Karolinska Institutet Foundation Grants, Petrus och Augusta Hedlunds Stiftelse, and The Glaucoma Foundation. PAW is an Alcon Research Institute Young Investigator.

## Author contributions
J.R.T. Performed experiments, performed analysis, created data visualization, wrote the manuscript, M.J. Performed experiments, performed analysis, created data visualization, wrote the manuscript, C.V. Performed experiments, performed analysis, created data visualization, wrote the manuscript, A.O. Performed experiments, A.C. Performed experiments, performed analysis, B.H. Performed experiments, E.D. Performed experiments, R.C. Performed analysis, A.L. Performed experiments, wrote the manuscript, J.G. Wrote the manuscript, C.E.W. Provided supervision, G.J. Provided supervision, R.C.B.W. Performed analysis, wrote the manuscript, M.P.C. Provided resources, wrote the manuscript, A.B. Provided resources, provided supervision, wrote the manuscript, conceptualized experiments/methodologies, P.A.W. Performed experiments, performed analysis, wrote the manuscript, conceptualized ideas/experiments/methodologies.

## Funding

## Competing interests
PAW is an inventor on an awarded US patent held by The Jackson Laboratory for nicotinamide treatment in glaucoma ("Treatment and prevention of ocular neurodegenerative disorder", US11389439B2). PAW, MJ, CV, and AB are inventors on a submitted patent held by Mim Neurosciences AB for novel NMNAT2-targeting small molecules. All other authors declare that they have no competing interests.
