## [Peer Review File · Nature Communications]

NMNAT2 is a druggable target to drive neuronal NAD productionREVIEWER COMMENTS

Reviewer #1 (Remarks to the Author):

This is an interesting manuscript by Tribble and colleagues, and is generally well written and presented. The authors show that the levels of a key neuronal NAD-producing enzyme, NMNAT2, is important for the survival of retinal ganglion cells (RGC). The NMNAT2 activator EGCG or NMNAT2 overexpression is shown to be neuroprotective in models of retinal neurodegeneration. The authors further used EGCG as a tool compound to develop various novel small molecules for driving NAD production in neurons. While the study is of basic biomedical relevance, substantial data is required to show the *in vivo* therapeutic efficacy of NMNAT2 activating small molecules, as well as additional neuroprotective properties, in order to be of general interest in this high-impact journal.

Major comments:

(1) In the context of the novelty of this study, it is important to demonstrate in an animal model the efficacy of the most potent NMNAT2 activating novel small molecule in preventing RGC degeneration and boosting NAD.

(2) The authors claim about designing novel neuron-specific NAD boosting small molecules. Show the effects of the most potent small molecules on NAD production in non-neuronal cells and tissues.

(3) The authors mention that mitochondrial impairment is a trigger for NMNAT2 depletion resulting in neurodegeneration. On the other hand, NAD is important for the function of mitochondrial electron transport chain. Show the effects of EGCG and a potent novel small molecule in improving mitochondrial membrane potential and respiration during RGC degeneration in glaucoma model.

(4) Autophagy plays a vital role in clearing protein aggregates and preventing neurodegeneration. Is there autophagy impairment and aggresome accumulation during neurodegeneration of RGCs in the models used in this study, and can the NMNAT2 activators rescue these phenotypes?

(5) A recent study has shown that autophagy regulates NAD levels, and NAD boosters are neuroprotective in autophagy-deficient neuronal cells (PMID: 37086404). Is it possible to test the neuroprotective effects of any of the potent NMNAT2 activating small molecule in this context?

(6) While there are bioavailable NAD supplements such as NMN and NR, write discussion about the utility of NMNAT2 activating novel small molecules for biomedical applications in comparison to existing strategies.

(7) Please refer to specific supplementary figure panels in the results text instead of referring to the whole figures while describing the results.

Reviewer #2 (Remarks to the Author):

In an interesting set of experiments by Tribble et al., authors propose a role for Nmnat2 as a modifiable treatment target in ocular neurodegeneration typified by glaucoma and began the process of engineered a drug to modify Nmnat2 activity/increase NAD production. The topic is interesting and significant, the data and methodology are sound, the manuscript could be strengthened by addressing the following:

Page 3 Main Text:

Line 17: Perhaps not surprising that Nmnat2 expression is variable across BXD strains as phenotypic variability including highly variable RGC numbers appear characteristic for this family of crosses.

Line 28: DBA/2J demonstrates a highly variable phenotype which affects IOP (which are not shown

here). I am concerned that a N of 4 is not sufficient to demonstrate IOP-dependence of Nmnat2 depletion in DBA/2Js as a glaucoma model. Increasing N here would be helpful.

Line 33: I am not certain that the conclusion "These data support a hypothesis in which low NMNAT2 leads to RGC degeneration or renders RGCs susceptible to neurodegenerative insults" is fully supported by the data shown in Figure 1A-E and Supplemental Figure 1. While later experiments do support this hypothesis, arriving at this conclusion at this point in the paper is premature. Recommend either altering or removing this sentence from this paragraph.

Line 48: I would be interested in seeing whether decreased Nmnat2 expression also affects RGC survival following OHT (as was shown following axotomy in Figure 1H). Compared to axotomy, this would more directly associate Nmnat2 levels with glaucoma susceptibility.

Page 4 Main text:

Line 14: Apart from regulatory hurdles, gene therapy is just much less likely to work for a multifactorial, multigenetic disease like glaucoma.

Line 21: Please indicate the degree of neuroprotection (% increase in RGC survival compared to untreated) following oral EGCG delivery in the rat OHT glaucoma model. This is difficult to discern based on figure 2C alone.

Line 35: I would be interested in seeing whether EGCG protects against OHT in addition to axotomy following NMN depletion, which more directly recapitulate glaucoma in animal models.

Page 5 Main text:

Paragraph 1: The authors have clearly done a lot of work identifying compound with better bioavailability and potency than EGCG. Inclusion of Figure 3 is of interest but could certainly be strengthened by going a step further to show effectiveness of one of these compounds in neuroprotection for axotomy and OHT models.

Minor:

Line 44: Recommend altering "This represents a combination the developmental loss of peripheral sensory axons and the late onset loss of motor axons previously identified in Nmnat2gtBay/gtE mice" to "This represents an ocular demonstration of combined developmental loss of peripheral sensory axons and late onset loss of motor axons previously identified in Nmnat2gtBay/gtE mice" or something similar if I have correctly interpreted the authors' meaning.

Line 49: Recommend changing "This suggests a threshold of Nmnat2 expression passed which RGCs are sensitized to neurodegenerative insults" to "This suggests a threshold for Nmnat2 expression past which RGCs are sensitized to neurodegenerative insults."

Reviewer #3 (Remarks to the Author):

In their manuscript entitled "NMNAT2 is a drugable target to drive neuronal NAD production", Tribble et al. describe a collection of data supporting the view that NMNAT2 is a therapeutic target for retinal ganglion cell (RGC) neuroprotection. First, they show the levels of this enzyme are variable in mice and humans, and high in RGCs relative to other retinal cells. By looking at mice with variable levels of NMNAT2, and by modulating the levels of the enzyme through viral expression of hNMNAT2, they show that the levels of this enzyme is correlated with RGC survival, during development and aging in mice but also after culturing the retinas, a model used to emulate axotomy. Most excitingly, they identify a polyphenol present in green tea, epigallocatechin gallate (EGCG), which appears to act by boosting the activity of NMNAT2, and which is neuroprotective in various of the assays shown to be sensitive to NMNAT2 levels. Then, they demonstrate that the effect of EGCG is inhibited by also inhibiting the NAMPT enzyme that makes the substrate for NMNAT2, NMN, supporting the view that EGCG is acting by regulating the NAD salvage pathway. Finally, they demonstrate that they can make rationally driven derivatives of EGCG most of which retain the NMNAT2 boosting activities. The work is highly significant because of its very large potential to lead to therapeutics for diseases affecting RGCs and other neurons.

There are, however, a number of concerns that raise into question whether the results presented fully support the conclusions.

The main concerns are:

1. Much of what was shown supporting NMNAT2 being neuroprotective has been previously shown, or at least not surprising given various other manipulations of NAD levels in RGCs.
2. That EGCG really acts by regulating NMNAT2 enzymatic activity is consistent with but not firmly established (as opposed to many of their effects being due to some other activity that ultimately leads to healthier cells and therefore more NAD). The strongest support comes from the use of an established NAMPT non-competitive inhibitor, but the effect is modest, and no biochemistry/enzymatic assays are provided to show conclusively that EGCG acts directly by modulating NMNAT2 enzymatic activity (non-competitively).
3. While amazingly promising, the derivatization of EGCG into more drugable compounds is underdeveloped. Indeed, one would want to demonstrate by biochemistry/enzymatics as well as neuroprotection assays that these drugs act as intended.

While some of these points may be addressable just by toning down the claims, to truly put them to rest would require additional experimentation.

Other concerns that, if corrected, might diminish these above concerns are:

1. It is not clear whether authors are claiming NMNAT2 levels vary more than other genes in both mice and human RGCs. It is inferred but not shown that the dose-dependent effect NMNAT2 matters more than for other "housekeeping" genes; that is not substantiated.
2. That NMNAT2 is higher in RGCs than other retinal cells and absent in glia is not truly supported. Nuclear seq is consistent but not conclusive, as RGCs have uniquely open chromatin and large nuclei that may behave differently to other cells when it comes to Nuclear Seq. Additional data (e.g., in situ hybridization or antibody labeling or the like) is needed to support this claim.
3. While useful, the RGC explant model is limited, as RGCs are rapidly dying, the culturing (and RGC survival) can be quite variable and variably affected by drug manipulations. In addition, the RGC quantification based on Rbpms is highly suspect if the whole retina is not quantified, due to large variations in RGC density along central to peripheral and dorsal to ventral axes. While the RGC explant model is OK to use, key findings really need to be validated by an in vivo ONC assay (and whole retina RGC counts or the equivalent).
4. The effect of FK866 is only partial. Would one not expect that the effect of EGCG to be absolutely dependent on NMN? This may be the case as FK866 may also only decrease NMN by 30% (in which case there is absolute dependence on NMN), but this would need to be demonstrated.
5. There also are conflicting data on how levels of NMNAT2 and EGCG interact. It should be the case that both NMNAT2 levels and EGCG cooperate at low and intermediate levels of NMNAT2 but that once NMNAT2 levels are maximal in terms of NAD production, that EGCG no longer has an effect. This is inferred but not demonstrated.
6. Since NMNAT2 is present in other neurons in the CNS, and likely other cells too, the data on NMNAT2 dosage affecting RGC survival during development, aging and after injury are somewhat oversold; the possibility of brain or even systemic effects need to be acknowledged.
7. The description of the derivatization of EGCG is hard to follow. For example, why was compound 18 the basis for further derivatization when there were more potent variables available. It also seems that all these details presented as results are really methods and not results, and that the actual results (how these new compounds behave in enzymatic vs neuroprotective assays) are not shown.
8. Not showing the actual chemical formulas of all the derivatives makes the science hard to evaluate, and makes the study impossible to reproduce. While IP makes sharing such data difficult, maybe the same conclusions could be made without disclosing the lead compounds (but disclosing others).

REVIEWER COMMENTS

Reviewer #1 (Remarks to the Author):

This is an interesting manuscript by Tribble and colleagues, and is generally well written and presented. The authors show that the levels of a key neuronal NAD-producing enzyme, NMNAT2, is important for the survival of retinal ganglion cells (RGC). The NMNAT2 activator EGCG or NMNAT2 overexpression is shown to be neuroprotective in models of retinal neurodegeneration. The authors further used EGCG as a tool compound to develop various novel small molecules for driving NAD production in neurons. While the study is of basic biomedical relevance, substantial data is required to show the *in vivo* therapeutic efficacy of NMNAT2 activating small molecules, as well as additional neuroprotective properties, in order to be of general interest in this high-impact journal.

The Authors would like to thank Reviewer #1 for their thoughtful comments with which to improve the current study. As the study was originally submitted in a short report / letter format, the Authors appreciate that some of the intricacies of the study may have been missed due to the short text format or 'lost' in the supplementary data. As such, we have expanded the manuscript into a full manuscript style and have significantly expanded the figures, Results, and Discussion to aid with comprehension of the study. Similar *in silico* or *in vitro* drug development projects within the NAD space (e.g. SARM1 inhibitors) have been well received in general interest journals (e.g. Hughes *et al.*, *Cell Reports*, Feldman *et al.*, *PNAS*) and with the suggested additions to this manuscript presented below we believe that this will now make a stronger, more attractive article.

Major comments:

(1) In the context of the novelty of this study, it is important to demonstrate in an animal model the efficacy of the most potent NMNAT2 activating novel small molecule in preventing RGC degeneration and boosting NAD.

The Authors agree that demonstrating a neuroprotective effect of these compounds would strengthen the manuscript and support the validity of the drug development work. To provide compelling evidence for neuroprotection in this manuscript, we have performed *ex vivo* proof of concept experiments with a selection of compounds in the retina axotomy model. We have now tested 5 of our top compounds based on NAD producing capacity and a favorable response to FK866. We selected compounds across the phases of the drug development (*i.e.* early iterations from group 2 and 3, and later iterations from groups 4 and 5). We demonstrate that 4 of these compounds provide a significant improvement to RGC survival over untreated controls at 3 days *ex vivo*. Two of these were partial protections (significantly greater than untreated controls, but also significantly less than naïve controls). Two compounds demonstrated robust protection (significantly greater than untreated controls, and also not significantly different to naïve controls). Taken together data provide a strong indication that our novel compounds drive NAD and provide neuroprotection, and that as we refine our chemistry the potential for neuroprotection increases. This data is now shown in **Figure 3G** and **Supplementary Figure 11A-B**.

The following has been added:

Results

“To explore the potential of these compounds to provide neuroprotection we tested 5 of these compounds in the retinal explant model. Two of these compounds (38, 56; Groups 2 and 3) demonstrated a significant, but incomplete, RGC survival compared to untreated controls (Supp Fig 11A-B) and two compounds (54, 55; Group 5) demonstrated a significant, almost complete, protection to RGCs following axotomy (Figure 3G). Taken together, we demonstrate that NMNAT2 is a druggable target to generate high levels of neuron-specific NAD and that these novel small molecules can provide neuroprotection in the retina.”

Discussion

“Importantly, these top compounds were able to provide neuroprotection against RGC degeneration ex vivo, supporting their utility and potential as neuroprotective therapeutics in the future. These novel small molecules serve as a starting point for further medicinal chemistry to develop NMNAT2-targeted drugs that could be used in vivo, and potentially progress into human clinical trials.”

The Authors agree that in an idealistic setting within a drug development / pharma company the next logical step is to perform a crucial proof-of-concept experiment in an animal model of glaucoma. However, we consider this lies beyond the scope of the current study because the current stage of this study is lead identification (*i.e.* we have identified the putative binding site and developed novel drug candidates predicted to bind to this region of the NMNAT2 protein). The next steps are lead identification (up to 1 year), *in vitro* ADME and toxicity (6-12 months), *in vivo* toxicity and PK/PD (6-12 months) (and potentially additional med chem during these stages); together these represent in excess of a 2-3 year timeframe and a multimillion dollar investment (as a best case scenario, also see: PMID: 34368939).

The most important factor here regarding this manuscript and this study is that these development milestones would need to be met in order to assess the correct route, dosages, and timeframes for a pre-clinical model as well as to apply for ethics to test these compounds *in vivo*. Any *in vivo* testing that one could perform now would not only be deemed unethical but would also enrich for false negatives (as a failure at this stage would be caveated by not knowing the ADMET and BBB permeability of the compound(s) in mouse and rat systems). As such, we consider this is an unfeasible request for the current study.

The retinal axotomy model is ideal for rapidly testing compounds as the model is *ex vivo* and there are no additional ethics to overcome. The model also benefits from keeping the retinal circuitry intact (as opposed to *in vitro* work) and the injury is uniform (damages all ~55,000 axons in the mouse optic nerve at the same place at the same time). The retinal axotomy model is also ideal in these settings as we have previously demonstrated that raising NAD and preventing Wallerian degeneration provide a robust neuroprotection in this model which has translated to mature pre-clinical models and even into the clinic; *e.g.* nicotinamide (axotomy model: PubMed ID 28209901, pre-clinical models: 28209901, 33932867, Phase II clinical trial: 32721104), the Wallerian degeneration slow allele – WldS (axotomy model: 36635457, pre-clinical models: 28487632), NMNAT2 gene therapy (axotomy model and pre-clinical model: this manuscript).

(2) The authors claim about designing novel neuron-specific NAD boosting small molecules. Show the effects of the most potent small molecules on NAD production in non-neuronal cells and tissues.

The premise for our “neuron-specific” comment comes from the fact that NMNAT2 is predominantly expressed in neurons (with a small amount of mRNA/protein present in hair follicles, the testis, and some endocrine tissues: www.proteinatlas.org/ENSG00000157064-NMNAT2/tissue). It is important to note that these are tissue profiles and as such will also

include some neuronal populations. We have also demonstrated that NMNAT2 is neuron-specific in the retina, and that this expression is predominantly in retinal ganglion cells (PMID: 36681854).

In addition to our neuron enriched tissue (cortex) we have now provided data for EGCG and several of the top compounds in neuron-low tissues (liver, spleen, skeletal muscle). It is important to note that these tissues will still contain some neurons (so we can't denote these are neuron-depleted tissues). We demonstrate that in these neuron-low tissues, EGCG does not increase NAD (**Supp Figure 3C**). We repeated this for the top 10 compounds based on NAD-producing capacity and demonstrate that a NAD boosting effect in neuron-low tissue was only demonstrated for 2 compounds (51, 56) with both increasing NAD in muscle by <1.2 fold relative to untreated controls. A reduction in NAD relative to control was identified for 3 compounds in neuron-low tissue, with compound 21 and 55 reducing NAD in spleen to ~0.8 fold, and compound 54 reducing NAD in muscle to ~0.9 fold (**Supp Figure 10C**).

The following has been added to the manuscript:

Results

"This effect is not replicated in neuron-low tissues (spleen, muscle, and liver; **Supp Figure 3C**)."

*"To test neuron specificity, NAD boosting effects of the top ten compounds were assessed in dissociated neuron-high (cortex) and neuron-low (liver, muscle, spleen) tissues. Supporting the previous findings with EGCG, the majority of the newly developed compounds retained neuron specificity (**Supp Fig 10C**)."*

Discussion

"Critically, EGCG increased NAD capacity in neuronal, but not non-neuronal, tissues."

(3) The authors mention that mitochondrial impairment is a trigger for NMNAT2 depletion resulting in neurodegeneration. On the other hand, NAD is important for the function of mitochondrial electron transport chain. Show the effects of EGCG and a potent novel small molecule in improving mitochondrial membrane potential and respiration during RGC degeneration in glaucoma model.

To assess whether EGCG could overcome mitochondrial impairment we assessed its neuroprotective capacity in three relevant models that demonstrate mitochondrial / metabolic impairment: *in vitro* rotenone-induced Complex I inhibition in dissociated retinal neuron culture, *in vivo* rotenone-induced Complex I inhibition, and *in vivo* rat ocular hypertension (which we have demonstrated to have retinal metabolic dysfunction early in the disease cascade: PMID: 33932867) (**Supp Figure 4A-B; Figure 2C**). We have now added to this by demonstrating that EGCG significantly changes mitochondrial membrane potential in dissociated cortical neurons at the concentrations previously tested in the above experiments (**Supp Figure 4C**). We also demonstrate that the majority of the top compounds alter mitochondrial membrane potential to a lesser extent than EGCG alone (**Supp Figure 10B**).

The following has been added:

Results

*"Cortical neurons demonstrated a significant increase in mitochondria membrane potential (via JC-1 staining) when treated with 5 μ M EGCG and a significant reduction in membrane potential at 50 μ M (**Supp Figure 4C**)."*

“Mitochondrial membrane potential was also assessed in these top compounds (Supp Fig 10B).”

Discussion

“...further supporting development towards an NMNAT2 specific mechanism of action and loss of other potential mechanisms (e.g. reduced effects on mitochondrial membrane potential compared to EGCG).”

(4) Autophagy plays a vital role in clearing protein aggregates and preventing neurodegeneration. Is there autophagy impairment and aggresome accumulation during neurodegeneration of RGCs in the models used in this study, and can the NMNAT2 activators rescue these phenotypes?

The potential role of autophagy in glaucoma is an area of recent interest (Villarejo-Zori *et al.*, 2021 PMID: 34620506). Induction of autophagy with rapamycin after optic nerve injury (23521856; 31409770) and in chronic glaucoma models has demonstrated neuroprotection in some cases but not in others (*i.e.* highly context-, time-, and dose- dependent) (24923557; 22476098; 33318177). In the specific models used in this study, autophagy has not been explored. Glaucoma is a complex neurodegenerative disease with many interacting mechanisms. Whilst autophagy could well play a role in the neurodegeneration of retinal ganglion cells, it is outside the scope of the current study to explore this mechanisms of neurodegeneration in glaucoma.

The following has been added to the Discussion:

“As a polyphenol, EGCG has a number of potential mechanisms of action that could be providing neuroprotection including through modulation of reactive oxygen species, cell signalling, growth factors, autophagy, and apoptotic cascades (43). Modulation of these could also affect downstream NAD levels (44). Glaucoma is a complex neurodegenerative disease in which these factors have previously been identified as potential pathological mechanisms (45). While EGCG did not increase NAD in non-neuronal tissues, this does not preclude other effects derived from these other properties if given as an oral treatment.”

(5) A recent study has shown that autophagy regulates NAD levels, and NAD boosters are neuroprotective in autophagy-deficient neuronal cells (PMID: 37086404). Is it possible to test the neuroprotective effects of any of the potent NMNAT2 activating small molecule in this context?

We thank the Reviewer for bringing this interesting paper to our attention. ATG5^{-/-} hESCs could be a future series of experiments to examine the potential of these compounds in an autophagy-deficient context. However, it is worth noting that in these published studies there is depletion of several metabolites related to glycolysis and tricarboxylic acid cycle, nucleotide energy carriers, and various amino acids in addition to NAD. As such, any experiment with our novel compounds would be highly caveated as we would not be able to confirm a mechanism of action. As such these experiments should be considered out of the scope of these studies and are better suited to a stand-alone series of experiments.

This paper has been cited in the Discussion (ref 44):

“As a polyphenol, EGCG has a number of potential mechanisms of action that could be providing neuroprotection including through modulation of reactive oxygen species, cell signalling, growth factors, autophagy, and apoptotic cascades (43). Modulation of these could

also affect downstream NAD levels (44). Glaucoma is a complex neurodegenerative disease in which these factors have previously been identified as potential pathological mechanisms (45). While EGCG did not increase NAD in non-neuronal tissues, this does not preclude other effects derived from these other properties if given as an oral treatment.”

(6) While there are bioavailable NAD supplements such as NMN and NR, write discussion about the utility of NMNAT2 activating novel small molecules for biomedical applications in comparison to existing strategies.

Our original manuscript was in the short report / letter format, and as such, much of this important discussion was cut for brevity. We have now reformatted the text as a full manuscript and, as such, we have more space and scope to discuss important points as astutely mentioned here.

The following has been added to the Discussion:

“A number of NAD precursors have been explored for their potential to increase NAD across different tissues, particularly the CNS where depletion of NAD pools drives, or contributes to, neurodegeneration. Nicotinamide has strong therapeutic potential given its long clinical history, strong neuroprotection in animal models of glaucoma and other neurodegenerative diseases (30), and functional benefits established in short-term clinical trials (31, 32). However, nicotinamide requires high doses to achieve meaningful NAD increases in the CNS and is not specific to neurons. Whilst nicotinamide riboside (NR) can achieve greater (although more transient) NAD increases than nicotinamide at lower doses (33), its utility in the retina is more limited due to the low expression of NRK (required to convert NR to NMN (6)) which produces only a modest 10-20% NAD increase (34). Stabilized versions of NMN, while able to raise NAD, are potentially dangerous alternatives in NAD depleted systems (such as in neurodegenerative diseases) given that high ratios of NMN to NAD trigger SARM1 activation and axon degeneration (35-37). Similarly, SARM1 inhibition through newly developed small molecule inhibitors are strong neuroprotective candidates but are reliant on large pools of NAD (38), which are absent during the neurodegenerative disease cascades. While SARM1 inhibitors will limit NAD consumption, they do not increase new NAD formation and so their long-term use may be limited. Supporting this, Sarm1^{-/-} does not prevent RGC soma loss following optic nerve crush (39). In the axon degeneration cascade, NMNAT2 acts up-stream of SARM1 (40), and so offers an alternate target, particularly as restoring the capacity of NMNAT2 activity towards normal levels is unlikely to present significant side effect in comparison to removal of a native function (e.g. blocking SARM1). In addition, inhibition of SARM1 could have deleterious effects given its protective role in preventing viral spread (41, 42). Novel small molecules that target NMNAT2 could provide neuronal targeted increases in NAD to boost NAD pools and prevent the initiation of neurodegenerative cascades.”

(7) Please refer to specific supplementary figure panels in the results text instead of referring to the whole figures while describing the results.

We have expanded the text to call out individual panels where appropriate.

Reviewer #2 (Remarks to the Author):

In an interesting set of experiments by Tribble et al., authors propose a role for *Nmnat2* as a modifiable treatment target in ocular neurodegeneration typified by glaucoma and began the process of engineered a drug to modify *Nmnat2* activity/increase NAD production. The topic is interesting and significant, the data and methodology are sound, the manuscript could be strengthened by addressing the following:

We thanks the Reviewer for their optimism and praise of our studies, and for the constructive criticism with which to improve the manuscript.

Page 3 Main Text:

Line 17: Perhaps not surprising that *Nmnat2* expression is variable across BXD strains as phenotypic variability including highly variable RGC numbers appear characteristic for this family of crosses.

The Reviewer raises an important consideration. Williams *et al.* 1998 (PMID: 9412494) did indeed demonstrate that total retinal ganglion cell numbers are variable across BXD individuals, however they identified a clear bimodal distribution in retinal ganglion cell density corresponding to the mean peaks of retinal ganglion cell density of the parental C57BL/6J (~55,000 retinal ganglion cells) and DBA/2J (~63,000 retinal ganglion cells) strains. Our retinal *Nmnat2* expression data shows a broad expression curve without a clear bimodal distribution. To further confirm this, we correlated mean *Nmnat2* expression to mean retinal ganglion cell density for each BXD strain using data from Williams *et al.* (PMID: 9412494) and retinal *Nmnat2* data from GeneNetwork (DoD CDMRP Retina Affy MoGene 2.0 ST (May15) RMA Gene and Exon Level). Spearman's rank correlation demonstrated that there was no correlation between *Nmnat2* expression and total RGC number ($r = -0.00088$, $P = 0.997$) demonstrating that the variability of *Nmnat2* expression across BXD strains is unlikely due to the variability in RGC numbers across these strains. This data has been added to Figure 1.

The following has been added:

Methods:

*"We determined whether *Nmnat2* expression was related to RGC variability by performing a Spearman's rank correlation test comparing the average *Nmnat2* expression in the retina from GeneNetwork (DoD CDMRP Retina Affy MoGene 2.0 ST (May15) RMA Gene and Exon Level) to the average RGC density of each strain from Williams *et al.*, (11)."*

Results:

*"In the retina, this variability is not related to the number of RGCs (Spearman's rank correlation $r = -0.00088$, $P = 0.997$; **Figure 1B**) which is known to vary across individual mice and strains (11)."*

Line 28: DBA/2J demonstrates a highly variable phenotype which affects IOP (which are not shown here). I am concerned that a N of 4 is not sufficient to demonstrate IOP-dependance of *Nmnat2* depletion in DBA/2Js as a glaucoma model. Increasing N here would be helpful.

We apologise for the lack of clarity regarding these data when presented in the short format. We used publicly available RNA-sequencing data from Williams *et al.*, 2017 (PMID:

28209901). In these experiments, the number of DBA/2J is indeed 4, but these are strictly phenotyped (all have no detectable neurodegeneration in the optic nerve) and so are not a variable sample. To further strengthen this, we have added further additional data from this dataset and another publicly available DBA/2J dataset (microarray gene-expression) with a higher number of samples ($n = 4-10$ / group).

We have added the following:

Methods:

“Nmnat2 expression across glaucoma progression was determined from publicly available microarray gene-expression data from whole optic nerve head (ONH) and whole retina in 10.5 month-old DBA/2J mice (12, 13). We explored Nmant2 expression across the disease clusters identified by Howell et al. (12) which are based on the degree of genetic change (and correspond to the degree of neurodegeneration in histological optic nerve analysis). In the ONH, these grouping are: Group 1 = no detectable glaucoma, limited genetic change, ($n = 8$); Group 2 = no detectable glaucoma, significant genetic change, ($n = 8$); Group 3 = no or early glaucoma, significant genetic change, ($n = 6$); Group 4 = moderate glaucomatous degeneration, significant genetic change, ($n = 4$); Group 5 = severe glaucomatous degeneration, significant genetic change, ($n = 4$). Expression is compared to $n = 5$ D2-Gpnmb+. In the retina, there are 4 groups: Group 1 = no detectable glaucoma, limited genetic change, ($n = 8$); Group 2 = no detectable glaucoma, moderate genetic change, ($n = 9$); Group 3 = moderate glaucomatous degeneration, significant genetic change, ($n = 3$); Group 4 = severe glaucomatous degeneration, significant genetic change ($n = 10$). Expression is compared to $n = 8$ D2-Gpnmb+. Nmnat2 expression in retinal ganglion cells early in glaucoma (9-month old DBA/2J) was explored using publicly available data from Williams et al. (3); all mice used for these experiments were confirmed to have no detectable neurodegeneration in the optic nerve ($n = 9$ D2-Gpnmb+ and $n = 26$ DBA/2J). Grouping represents degree of genetic change (1 = least, 4 = most): Group 1 ($n = 9$ age-matched D2-Gpnmb+ and $n = 6$ DBA/2Js), Group 2 ($n = 6$ DBA/2Js), Group 3 ($n = 10$ DBA/2Js), Group 4 ($n = 4$ DBA/2Js).”

Results:

*“In the D2 mouse (a model of a complex age- and IOP- dependent inherited glaucoma), Nmnat2 expression (as assessed by whole tissue microarray (12, 13)) is significantly reduced prior to detectable neurodegeneration in the ONH (molecular disease group 3, $P = 0.025$) and continues to decline in moderate and severe disease (molecular disease group 4, $P = 0.001$, and 5, $P < 0.001$; **Figure 1F**). In the retina, Nmnat2 is significantly reduced in severe disease (molecular disease group 4, $P < 0.001$; **Figure 1F**). However, when only considering RGCs (FAC sorted, bulk RNA sequencing (3)), Nmnat2 is significantly decreased at a time point prior to detectable neurodegeneration in eyes with the greatest degree of transcriptional dysfunction (molecular disease group 4, $P = 0.004$; **Figure 1F**).”*

For additional background information, we summarize these experiments below:

We used publicly available RNA-sequencing data from Williams et al., 2017 (PMID: 28209901). In these experiments, retinal ganglion cells were isolated by FACS for bulk RNA-sequencing. This was performed in 9 month old mice which, in that colony, represent a timepoint at which IOP is high, but there is no detectable neurodegeneration. All 9-month DBA/2J mice used for those experiments were confirmed to have no detectable

neurodegeneration in the optic nerve (as assessed by PPD staining and axon counts: 28209901 and 29497468). However, as the reviewer correctly states, the onset and magnitude of IOP is highly variable in DBA/2Js. As such, mice may have greater cumulative exposure to IOP despite not showing signs of neurodegeneration. As such, the eyes can present with differing degrees of molecular changes despite having no detectable neurodegeneration. Sequencing efforts in the DBA/2J have therefore sought to control for this variability by employing statistical clustering methods to large numbers of samples to group eyes by the degree of molecular change (PMID: 21383504; PMID: 30670050; PMID: 32450896). This has previously been used to identify and validate molecular pathways and mechanisms for neurodegeneration and neuroprotection (PMID: 21383504). In Williams *et al.*, 2017 (PMID: 28209901), $n = 35$ individual retinas were grouped in this manner using unbiased hierarchical clustering to give 4 distinct, molecularly defined groups. ($n = 9$ D2-*Gpnmb*⁺ controls and $n = 26$ DBA/2J, all 9 months old)

- Group 1 (“least” disease) was a cluster containing all controls ($n = 9$; age-matched D2-*Gpnmb*⁺, which do not develop high IOP or neurodegeneration) and $n = 6$ DBA/2Js.
- Group 2 was a cluster containing $n = 6$ DBA/2Js
- Group 3 was a cluster containing $n = 10$ DBA/2Js
- Group 4 (“most disease”) was a cluster containing $n = 4$ DBA/2Js (the open access data we reused in the original version of this manuscript).

(*N.B.* It is important to note that all optic nerves were identical for these mice and therefore “least” and “most” disease refer to a molecular grade only.)

These groups represent the variability of IOP-related stress and demonstrate increasing dysregulation of mRNA transcript. It is important to re-iterate, that all of these have no detectable neurodegeneration. We initially only plotted *Nmnat2* expression in Group 1 and Group 4. The intent was to avoid complicating a manuscript with the text limitations of a *Nature* letter with this nuance. We agree that this had introduced unnecessary confusion over the data. We have now included all DBA/2J retinas (Groups 1-4). Furthermore, we have included additional publicly available microarray gene-expression data from whole optic nerve head and whole retina in 10.5 month-old DBA/2J mice (PMID: 21383504). These data are clustered based on the degree of genetic change and this corresponds to the degree of neurodegeneration in histological optic nerve analysis:

- Group 1 = no detectable glaucoma, limited genetic change, $n = 8$
- Group 2 = no detectable glaucoma, significant genetic change, $n = 8$
- Group 3 = no or early glaucoma, significant genetic change, $n = 6$
- Group 4 = moderate glaucomatous degeneration, significant genetic change, $n = 4$
- Group 5 = severe glaucomatous degeneration, significant genetic change, $n = 4$).

In the retina, clustering identified 4 groups:

- Group 1 = no detectable glaucoma, limited genetic change, $n = 8$
- Group 2 = no detectable glaucoma, moderate genetic change, $n = 9$
- Group 3 = moderate glaucomatous degeneration, significant genetic change, $n = 3$
- Group 4 = severe glaucomatous degeneration, significant genetic change, $n = 10$).

Exploration of these data supported our initial findings of *Nmnat2* decline in the DBA/2J, with *Nmnat2* significantly decreased in whole ONH (enriched for retinal ganglion cell axons) beginning at an early time point (Group 3) and progressing with disease. In whole retina, this was detected only in severe disease. These data are shown in Figure 1F.

Line 33: I am not certain that the conclusion “These data support a hypothesis in which low NMNAT2 leads to RGC degeneration or renders RGCs susceptible to neurodegenerative insults” is fully supported by the data shown in Figure 1A-E and Supplemental Figure 1. While later experiments do support this hypothesis, arriving at this conclusion at this point in the paper is premature. Recommend either altering or removing this sentence from this paragraph.

We have altered this sentence to the following:

“We therefore hypothesized that low NMNAT2 leads to RGC degeneration or renders RGCs susceptible to neurodegenerative insults. To test this, we used mice carrying one or two, fully or semi-penetrant, gene trapped (gt) alleles for Nmnat2 to titrate Nmnat2 levels in the retina (Figure 1H).”

Line 48: I would be interested in seeing whether decreased Nmnat2 expression also affects RGC survival following OHT (as was shown following axotomy in Figure 1H). Compared to axotomy, this would more directly associate Nmnat2 levels with glaucoma susceptibility.

Testing whether rats with decreased *Nmnat2* have worse retinal ganglion cell survival in glaucoma would indeed be a valuable experiment to associate *Nmnat2* levels with glaucoma susceptibility. However, there are several technical and biological barriers to performing these experiments. No rat strains analogous to the *Nmnat2*^{gtBay/gtE} mice exist. Alternatively, reducing *Nmnat2* with shRNA/siRNA or similar is caveated by evidence of spontaneous neurite degeneration following these interventions (PMID: 20126265, PMID: 30304512). In addition, commercial siRNAs only provide ~60% *Nmnat2* knockdown *in vivo* which would add a layer of complexity in analyzing data generated (as the data we present in this manuscript suggest a critical threshold of a least <25% *Nmnat2* (**Figure 1J**)). Together these would not faithfully represent the decline of *Nmnat2* that we have identified in glaucoma models and would be highly likely to lead to spontaneous retinal ganglion cell drop-out prior ocular hypertension-induction. In contrast, since *Nmnat2*^{gtBay/gtE} mice exist with low *Nmnat2* from birth, this does not trigger axon degenerative mechanisms until neuronal stress is applied (e.g. age or injury) as we demonstrate in this manuscript and as has been previously demonstrated (PMID: 20126265, PMID: 30304512, PMID: 23946398).

Page 4 Main text:

Line 14: Apart from regulatory hurdles, gene therapy is just much less likely to work for a multifactorial, multigenetic disease like glaucoma.

We have edited to the text to include this point:

“Whilst gene therapy has found success in treating monogenic eye diseases, it is less likely to provide complete protection for complex polygenic neurodegenerative diseases, such as glaucoma, and its licensed use is likely to face many regulatory hurdles.”

Line 21: Please indicate the degree of neuroprotection (% increase in RGC survival compared to untreated) following oral EGCG delivery in the rat OHT glaucoma model. This is difficult to discern based on figure 2C alone.

We have added the mean % loss from control / mean % RGC survival relative to OHT untreated to each condition in Figure 2C to aid in readability.

Line 35: I would be interested in seeing whether EGCG protects against OHT in addition to axotomy following NMN depletion, which more directly recapitulate glaucoma in animal models.

We demonstrate that EGCG protects against OHT (**Figure 2C**) and axotomy (**Figure 2C**) and that this protective effect is blocked in axotomy when FK866 is present (**Figure 2E**). Combining FK866 with OHT would face several technical and biological barriers.

1. FK866 would have to be delivered by intravitreal injection to avoid systemic metabolic dysfunction. The concentrations and timeframes for such an experiment are not known and testing these without prior information would carry significant ethical risk. To fully block NMN in the retina would require a careful titration of the FK866 dose (and duration of effect) that would require targeted MS to definitively assess NMN levels to establish a retina specific FK866 protocol *in vivo*.
2. FK866 is not neuron specific. Since FK866 works upstream of Nmnats, by inhibiting Nampt, the effects of this would occur in all nucleated cells of the eye. In the explant, this is less of a concern since the experiment time scale is rapid (3 days) and other retinal neurons are resistant to neurodegenerative effects over this time scale (PMID: 21345987). The OHT model is a 2-week model and the potential for off-target effects are greater. Genetic Nmnat1 deficiency (e.g. p.Val9Met mutation) manifest as early-onset retinal degeneration through photoreceptor degeneration, on a similar time scale to the duration of the OHT model (PMID: 37214313, PMID: 34750622). Thus, any loss of Nmnat1 activity in the retina *in vivo* would significantly caveat any findings. Similarly, knockout of *Nampt* (which would be roughly equivalent to Nampt inhibition by FK866) results in an almost complete loss of the outer retina (PMID: 27681422). This would further caveat any data generated.
3. The temporal dynamics of NMN in glaucoma have yet to be determined although it is highly likely that NMN increases in OHT (through increased NAD consumption (PMID: 34198948)) with progressive neurodegeneration. To block NMN production in OHT would therefore require increasing doses of FK866 to counteract new production of NMN and maintain a blockade of Nmnat2 activity. This would require multiple, repeated, intravitreal injections which would add additional caveats to the model. In addition, there will still be residual cellular NAD that has yet to be consumed which adds an additional caveat.
4. NAMPT is not the only enzyme that makes NMN, but it is the predominant one in neurons, therefore we cannot rule out non-NAMPT routes to NMN, e.g. via NR. This is limited in the retinal explant model, but in the OHT model would be a greater consideration since we know that non-retinal cells in the eye have high expression of NMRK1 (NR to NMN), particularly ciliary body cells (PMID: 36681854), which could release NMN in to the vitreous, as well as systemic routes to metabolism contributing to NMN / NAD levels in the eye.

Page 5 Main text:

Paragraph 1: The authors have clearly done a lot of work identifying compound with better bioavailability and potency than EGCG. Inclusion of Figure 3 is of interest but could certainly be strengthened by going a step further to show effectiveness of one of these compounds in neuroprotection for axotomy and OHT models.

Please see the response to Reviewer#1, comment #1.

Minor:

Line 44: Recommend altering “This represents a combination the developmental loss of peripheral sensory axons and the late onset loss of motor axons previously identified in *Nmnat2^{gtBay/gtE}* mice” to “This represents an ocular demonstration of combined developmental loss of peripheral sensory axons and late onset loss of motor axons previously identified in *Nmnat2^{gtBay/gtE}* mice” or something similar if I have correctly interpreted the authors’ meaning.

We thank for the Reviewer for providing a suggested sentence with better clarity. We have edited this sentence as suggested to:

*“This represents an ocular demonstration recapitulating a combination of the developmental loss of peripheral sensory axons and the late onset loss of motor axons previously identified in *Nmnat2^{gtBay/gtE}* mice.”*

Line 49: Recommend changing “This suggests a threshold of *Nmnat2* expression passed which RGCs are sensitized to neurodegenerative insults” to “This suggests a threshold for *Nmnat2* expression past which RGCs are sensitized to neurodegenerative insults.”

Typo corrected as suggested.

Reviewer #3 (Remarks to the Author):

In their manuscript entitled “NMNAT2 is a drugable target to drive neuronal NAD production”, Tribble et al. describe a collection of data supporting the view that NMNAT2 is a therapeutic target for retinal ganglion cell (RGC) neuroprotection. First, they show the levels of this enzyme are variable in mice and humans, and high in RGCs relative to other retinal cells. By looking at mice with variable levels of NMNAT2, and by modulating the levels of the enzyme through viral expression of hNMNAT2, they show that the levels of this enzyme is correlated with RGC survival, during development and aging in mice but also after culturing the retinas, a model used to emulate axotomy. Most excitingly, they identify a polyphenol present in green tea, epigallocatechin gallate (EGCG), which appears to act by boosting the activity of NMNAT2, and which is neuroprotective in various of the assays shown to be sensitive to NMNAT2 levels. Then, they demonstrate that the effect of EGCG is inhibited by also inhibiting the NAMPT enzyme that makes the substrate for NMNAT2, NMN, supporting the view that EGCG is acting by regulating the NAD salvage pathway. Finally, they demonstrate that they can make rationally driven derivatives of EGCG most of which retain the NMNAT2 boosting activities. The work is highly significant because of its very large potential to lead to therapeutics for diseases affecting RGCs and other neurons. There are, however, a number of concerns that raise into question whether the results presented fully support the conclusions.

We would like to thank the Reviewer for their review of this manuscript and praise of the studies within it.

The main concerns are:

1. Much of what was shown supporting NMNAT2 being neuroprotective has been previously shown, or at least not surprising given various other manipulations of NAD levels in RGCs.

The critical role of *Nmnat2* in axon/Wallerian degeneration and neuroprotection has been well established across multiple neuronal cell types and systems. In the retina, and in glaucoma, the relationship of *Nmnat2* to neurodegeneration and neuroprotection has yet to be well explored, but if properly analysed has the potential to be transformative for ophthalmic neurodegenerative diseases. As the retina is becoming more and more recognised in neurodegenerative diseases such as Alzheimer’s disease, the findings here could be more widely applicable and add to the scope and importance of our findings presented here (especially in monitoring treatment efficacy through retinal imaging in other neurodegenerative diseases).

We were the first to identify declining NAD as a critical, and treatable, component of early glaucoma pathophysiology. In the DBA/2J mouse we identified that NAD declined in the retina, and that repleting NAD either by supplementing with high doses of nicotinamide in the diet, or by gene therapy with *Nmnat1* (to enhance somal production of NAD), could prevent retinal NAD decline and provide a robust neuroprotection (Williams *et al.*, 2017 PMID: 28209901). We subsequently demonstrated declining NAD and neuroprotection with nicotinamide in a number of retinal ganglion cell injury and glaucoma models, which translated directly to demonstrate its potential to improve visual function in two Phase II clinical trials (Tribble *et al.*, 2021 PMID: 33932867; Hui *et al.*, 2020 PMID: 32721104; De Moraes *et al.*, 2022 PMID: 34792559). In the DBA/2J mouse we identified that *Nmnat2* declined prior to detectable

neurodegeneration, but this avenue was not explored further (Williams *et al.*, 2017 PMID: 28209901; Supp Fig 7F). Subsequently, sequencing of translating mRNAs isolated from retinal ganglion cell ribosomes demonstrated that *Nmnat2* was significantly downregulated in a mouse silicone oil model of OHT (Fang *et al.*, 2022 PMID: 35114390). Critically, this was during degenerative events (mean 80% axon survival, 60% peripheral retinal ganglion cell soma survival with a large standard deviation) and so the loss of *Nmnat2* is caveated by the loss of retinal ganglion cell density (as *Nmnat2* is enriched in retinal ganglion cells, please also see our comments above and PMID: 36681854, Figure 4 demonstrating a significant loss of NMNAT2 in glaucoma patients with a severe disease status). Given the importance of *Nmnat2* and NAD to axon and cell survival, exploring the role of *Nmnat2* in glaucoma is a logical advance. The intent of this article is not to portray this as a conceptual advance, but these initial hNMNAT2 gene therapy experiments are important as it provides a proof-of-concept that raising NMNAT2 is strongly neuroprotective which is of importance to our later drug development work. As such we have revised the introduction to better explain this rational.

Introduction:

“We previously identified downregulation of Nmnat2 occurring in RGCs prior to neurodegeneration in the DBA/2J mouse model of glaucoma (3, 8). This was subsequently supported by sequencing of translating mRNAs isolated from RGC ribosomes at a degenerative timepoint (where RGC loss has occurred) in a mouse ocular hypertensive (OHT) model (9). Similarly, we have demonstrated that NMNAT2 immuno-labelling is decreased in late-stage glaucoma in human retina and optic nerve head (ONH; a critical site of injury to RGC axons), where substantial RGC death has occurred (6). Given the critical role that NMNAT2 plays in axon maintenance and degeneration, understanding how NMNAT2 levels may influence RGC degeneration is of particular importance. Whilst genetic targeting of NMNAT2 been demonstrated to be robustly neuroprotective in other neuronal systems, there are no identified drugs or compounds that target endogenous NMNAT2 to produce high levels of NAD in neurons. We identify that epigallocatechin gallate (EGCG) drives NAD production in neurons through an NMNAT2 dependent mechanism. Using EGCG as a tool compound we develop the first small molecules driving neuronal NAD production through NMNAT2 which can also provide neuroprotection against RGC injury.”

The *Nmnat2* data is conceptually important to this manuscript and to the understanding of glaucoma because we identify the following novel findings:

1. We demonstrate that *Nmnat2* expression is overwhelmingly restricted to retinal ganglion cells in the retina, and that its expression is highly variable across individuals from mice to humans (and across individual retinal ganglion cells) (**Figure 1A-E**).
2. We demonstrate that *Nmnat2* declines in response to isolated features of glaucoma – high IOP and axon injury (**Figure 1G**).
3. We demonstrate that titrated loss of *Nmnat2* worsens retinal ganglion cell survival during development, with age, and with injury when *Nmnat2* expression drops below a threshold. (**Figure 1H-J**). This supports that *Nmnat2* loss in glaucoma is an important component of neurodegeneration and that individuals on the lower spectrum of *Nmnat2* expression may be more susceptible to retinal ganglion cell loss. This is important as we demonstrate a large spectrum of NMNAT2 levels in humans (both between individuals and across individual RGCs) (**Figure 1D-E**).

4. We demonstrate that restoring *Nmnat2* expression in depleted systems, via gene therapy, removes this susceptibility and provides neuroprotection (**Figure 1K**).

5. We also demonstrate that overexpression of the human isoform of NMNAT2 (hNMNAT2) via gene therapy is neuroprotective in retinal ganglion cell injury and glaucoma models (**Supp Figure 2D-E**). Overexpression of *Nmnat2* to provide neuroprotection has been used across multiple neuronal cell types and systems. We use hNMNAT2 overexpression via gene therapy as a tool to confirm that *Nmnat2* levels impact retinal ganglion cell survival in glaucoma (as mentioned above). We were the first to demonstrate this given that this has been published as a meeting abstract (PA Williams *et al.*, Targeting NMNAT2 for neuroprotection in glaucoma. *Invest. Ophthalmol. Vis. Sci.* 2022;63(7):1134).

Fang *et al.*, (2022) have also demonstrated that increasing *Nmnat2* stability via gene therapy is neuroprotective in glaucoma. In these experiments Fang and colleagues use a construct containing *Nmnat2*Δex6. Critically, this is a mouse *Nmnat2* which lacks the domain necessary for palmitoylation and membrane attachment. This drives *Nmnat2* into the cytosol and significantly increases its protein stability, reduces levels of ubiquitination, and provides a strong neuroprotection in cell culture which is stronger than native *Nmnat2* overexpression or Wld^S (PMID: 23610559). In this regard, *Nmnat2*Δex6 behaves functionally similar to the Wld^S fusion protein (PMID: 23610559) which has previously been demonstrated to be robustly neuroprotective in glaucoma (PMID: 28487632, PMID: 18158332, PMID: 34090501). Increasing the stability of *Nmnat2* or altering the localization of *Nmnat2*s has previously been demonstrated to provide robust neuroprotection (PMID: 23610559, PMID: 23995269, PMID: 30150401).

Our *Nmnat2* experiments establish the proof of concept that *Nmnat2* is important in glaucoma. As the Reviewer rightly comments, this is unsurprising, yet evidence supporting this assumption is nonetheless important (and, until this manuscript, had not been demonstrated elsewhere). Importantly we use this as a proof of concept to build a foundation to explore alternative routes to modulating *Nmnat2* for neuroprotection.

The following has been added to the Discussion:

*“Our study demonstrates that NMNAT2 expression is overwhelmingly restricted to RGCs in the retina, is highly variable within RGCs and across individuals, and that when expression falls below a critical threshold, RGC degeneration is significantly compounded. This supports NMNAT2 loss (RNA and/or protein) in glaucoma as an important component of neurodegeneration and that individuals on the lower spectrum of NMNAT2 expression may be more susceptible to RGC loss. Similarly, NMNAT2 expression (RNA and protein) is highly variable in aged postmortem human brains, and further decreased in brains with Alzheimer’s disease where lower NMNAT2 is correlated with worse cognition (10). We demonstrate that restoring NMNAT2 expression in depleted retinas via gene therapy removes this susceptibility to RGC degeneration and provides neuroprotection. Neuroprotection through similar mechanisms has previously been demonstrated in other neuronal systems through expression of WldS (the protein product of which functionally recapitulates the physiological role of NMNAT2 (14)) or through modulating the stability and subcellular localization of NMNAT2 (23-25). Similarly, Fang et al. (9) demonstrated that gene therapy delivery of *Nmnat2*Δex6, a more stable cytosolic form of *Nmnat2*, is also neuroprotective. In cell culture, *Nmnat2*Δex6 provides stronger neuroprotection than native *Nmnat2* overexpression or WldS (26). In this regard,*

Nmnat2 Δ ex6 behaves functionally similar to the WldS protein (26) which has previously been demonstrated to be robustly neuroprotective in glaucoma (27-29)."

2. That EGCG really acts by regulating NMNAT2 enzymatic activity is consistent with but not firmly established (as opposed to many of their effects being due to some other activity that ultimately leads to healthier cells and therefore more NAD). The strongest support comes from the use of an established NAMPT non-competitive inhibitor, but the effect is modest, and no biochemistry/enzymatic assays are provided to show conclusively that EGCG acts directly by modulating NMNAT2 enzymatic activity (non-competitively).

We agree that the current evidence is only supportive of EGCG acting directly on NMNAT2 with no experiment which tests this definitively. Our aim with EGCG was to use it as a tool compound to guide drug development (not to make a 'drug' version of EGCG). We demonstrate that FK866, the established NAMPT non-competitive inhibitor which blocks NMN production upstream of NMNAT2, significantly blocks the NAD producing capacity of EGCG (**Figure 2E, left panel**). In samples treated with EGCG and FK866 in combination, NAD is significantly reduced relative to EGCG only (-59%, $P = 0.002$; important to note that FK866 won't remove existing NAD, only preventing new NAD being produced), and is not significantly different to untreated controls ($P = 0.08$). Statistically the effect is complete. EGCG and FK866 in combination does give a statistically higher NAD concentration than FK866 only controls (+48%, $P = 0.003$). It is important to note that this difference is exaggerated by the fact that FK866 only samples have 22% lower NAD than controls. Since this is performed in cells rather than as a biochemistry/enzymatic assay, we cannot confirm that we have achieved a total depletion of NMN, and residual NMN and NAD will remain in the system (as only new production is blocked). The addition EGCG after 1 hour of FK866 could therefore have access to some residual substrate to make NAD. Longer incubation with FK866 could introduce confounders of cell viability and introduce a cell health effect as commented by the Reviewer.

In the absence of high quality NMNAT2 protein preparations and specific NMNAT2 inhibitors we cannot test this in an enzymatic assay. Supporting the interaction of EGCG and NMNAT2, we demonstrate that the survival benefit of EGCG to RGCs is significantly reduced in mice expressing <25% *Nmnat2* compared to WT (55% survival compared to 100% survival; **Figure 2F**), that *in silico* modelling of NMNAT2 predicts a binding site for EGCG, and that EGCG does not increase NAD in neuron-low tissues (liver, spleen, skeletal muscle; **Supp Figure 3C**). However, we recognize that these are not direct evidence, and have amended the manuscript accordingly. An important finding in this manuscript is that when the novel compounds are tested in the same FK866 assay (**Figure 3E**) we do demonstrate a complete loss of the NAD boosting effect in a number of the top 10 compounds, with comparable levels of NAD to FK866 treated controls, supporting the refinement of these compounds towards a greater specificity than demonstrated by EGCG.

The following has been added to the Discussion:

"Supporting that this occurs due to specificity for NMNAT2, EGCG's NAD boosting and neuroprotective effects are reduced or blocked in the presence of the NAMPT inhibitor FK866 or with Nmnat2 depletion. Our in silico data support the existence of a binding site for EGCG on an NMNAT2 homology model"

"Rather than develop a strategy for delivering therapeutic levels of EGCG (e.g. via encapsulation or direct targeting to the eye) we instead used EGCG as a tool compound to rationally design novel NMNAT2 targeting small molecules. We confirmed that our top

candidates significantly increase NAD capacity in an NMN dependent manner, supporting specificity to the NAD salvage pathway, whilst in silico modelling supported the maintenance of binding capability to NMNAT2. Compounds in the later groups of refinement (e.g. Group 4 and 5) demonstrated total loss of NAD boosting activity when co-incubated with FK866, more so than for EGCG, further supporting development towards an NMNAT2 specific mechanism of action and loss of other potential mechanisms (e.g. reduced effects on mitochondrial membrane potential compared to EGCG)."

3. While amazingly promising, the derivatization of EGCG into more drugable compounds is underdeveloped. Indeed, one would want to demonstrate by biochemistry/enzymatics as well as neuroprotection assays that these drugs act as intended.

Please see the Comment #1 from Reviewer #1 and the addition of neuroprotection data in **Figure 3G**.

While some of these points may be addressable just by toning down the claims, to truly put them to rest would require additional experimentation.

Thank you for the comments. We have now addressed all these major points with either adding additional supportive data, toning down the claims, or both. Please see the responses to individual comments.

Other concerns that, if corrected, might diminish these above concerns are:

1. It is not clear whether authors are claiming NMNAT2 levels vary more than other genes in both mice and human RGCs. It is inferred but not shown that the dose-dependent effect NMNAT2 matters more than for other "housekeeping" genes; that is not substantiated.

We agree that we had not provided sufficient evidence to claim that *Nmnat2* levels varied more than other genes in retinal ganglion cells. To address this, we first compared the variance in *Nmnat2* expression to the variance in established retinal ganglion cell-enriched genes in the mouse retina across BXD strains. Paired Pitman-Morgan tests demonstrated a significantly higher variance in *Nmnat2* expression than in *Pou4f1* (Brn3a), *Rbpms*, and *Tubb3* (β -III tubulin) supporting that *Nmnat2* is more variable than a number of retinal ganglion cell "housekeeping" genes. Data is now shown in Figure 1B. The following has been added:

Methods:

*"We determined whether *Nmnat2* expression was related to RGC variability by performing a Spearman's rank correlation test comparing the average *Nmnat2* expression in the retina from GeneNetwork (DoD CDMRP Retina Affy MoGene 2.0 ST (May15) RMA Gene and Exon Level) to the average RGC density of each strain from Williams et al., (11). From the same dataset, the variance in expression of *Nmnat2* was compared to RGC markers *Pou4f1*, *Rbpms*, *Thy1*, and *Tubb3* by performing Paired Pitman-Morgan tests using the *var.test* function in the PairedData package for R (<https://rdocumentation.org/packages/PairedData/versions/0.9.1>)."*

Results:

“Supporting this, Nmnat2 expression has a significantly greater variance than RGC markers Pou4f1 ($P < 0.001$), Rbpms ($P < 0.001$), and Tubb3 ($P = 0.006$), but not Thy1 ($P = 0.124$; Figure 1B).”

2. That NMNAT2 is higher in RGCs than other retinal cells and absent in glia is not truly supported. Nuclear seq is consistent but not conclusive, as RGCs have uniquely open chromatin and large nuclei that may behave differently to other cells when it comes to Nuclear Seq. Additional data (e.g., in situ hybridization or antibody labeling or the like) is needed to support this claim.

To address the potential caveat that retinal ganglion cells may show higher NMNAT2 in Nuclear Seq, we have now also queried publicly available single cell (sc)RNAseq data from human retina from Gautam et al. (2021 PMID: 34584087). scSeq supported the findings of nucSeq, also demonstrating that NMNAT2 expression is overwhelmingly constrained to retinal ganglion cell cells (**Figure 1E**). We have also used IHC in enucleated human eyes to demonstrate that NMNAT2 protein is not present in glial cells (PMID: 36681854). The following has been added:

Methods:

“NMNAT2 expression variability within individual RGCs was assessed using single-cell RNA-sequencing data from Gautam et al., (46) (GEO: GSE147979) and single-nucleus RNA-sequencing data from Orozco et al., (48) (GEO: GSE135133). The datasets were processed using Seurat V3.2. QC was performed to remove doublets and empty droplets by filtering cells with UMI 1000-6000 and detected genes with 500-5000. Clustering and cell annotation ID provided by the authors were used to identify the major cell types in the retina. To compare NMNAT2 expression across individual RGCs, the NormalizeData() function was used to generate normalized and log-transformed single cell expression with a scale factor of 10000. To compare NMNAT2 expression across all retinal cell types data were scaled using the ScaleData() function following identification of highly variable features using the FindVariableFeatures() function (vst method; 2000 features).”

Results:

*“Single-cell and single-nucleus RNA-sequencing from human retina confirms that retinal expression of NMNAT2 is highly variable across individual RGCs, which have the highest average expression among retinal neuronal (**Figure 1E**; NMNAT2 expression is not present in non-neuronal cell types in the retina).”*

3. While useful, the RGC explant model is limited, as RGCs are rapidly dying, the culturing (and RGC survival) can be quite variable and variably affected by drug manipulations. In addition, the RGC quantification based on Rbpms is highly suspect if the whole retina is not quantified, due to large variations in RGC density along central to peripheral and dorsal to ventral axes. While the RGC explant model is OK to use, key findings really need to be validated by an in vivo ONC assay (and whole retina RGC counts or the equivalent).

The retinal axotomy model is ideal for rapidly testing compounds as, in the mouse, retinal ganglion cell degeneration occurs within a 3-5 day window, and as the model is *ex vivo* tissue

there are no additional ethics to overcome. This makes the model an ideal first testbed for experiments. In our hands, we achieve a reliable degeneration of ~50% on average. While there can be variation in the death of RGCs at 3 days, we are powered to identify changes at 6 retinas. In a number of cases, the effects of our manipulations are almost binary (*i.e.* full or no neuroprotection, *e.g.* **Figure 3B, 3E, 3F**) demonstrating the robustness of this model. We have previously demonstrated that raising NAD and preventing Wallerian degeneration provide a robust neuroprotection in this model which has translated to mature pre-clinical models and even into the clinic; *e.g.* nicotinamide (axotomy model: PMID: 28209901, pre-clinical models: PMID: 28209901, 33932867, Phase II clinical trial: PMID: 32721104), the Wallerian degeneration slow allele – *Wld^S* (axotomy model: PMID: 36635457, pre-clinical models: PMID: 28487632), NMNAT2 gene therapy (axotomy model and pre-clinical model: this manuscript). We argue that this provides strong validity to initial testing in this model, especially in the context of NAD-related therapies (such as hNMNAT2 gene therapy, EGCG, NAM, and our novel compounds). We also support the EGCG and the NAM data by using retinal punches from human donor retinas which demonstrate similar survival profiles (**Figure 2D**), further validating this model's utility when looking at human disease.

We then proceeded further test and validate findings in an *in vivo* glaucoma model (**Figure 2C**) and an *in vivo* rotenone model (**Supp Figure 4B**) which both produce consistent loss of ~40% of RGCs. Here we identified that EGCG has some neuroprotective capacity, but that the effects were less robust than *ex vivo* which we attribute to the bioavailability of EGCG which is limited by GI tract pH (PMID: 10965518; not a factor in the *ex vivo* retinal explant model). We would question the benefits of performing a third *in vivo* model when we would expect to see a similar level of protection. We believe that this is ethically difficult to justify, especially since we do not consider EGCG to be a viable therapeutic, and next moved towards developing more drug-like novel compounds. It is also important to note that we performed our EGCG and hNMNAT2 gene therapy experiments in an OHT model by occlusion of the aqueous humor drainage structures in the eye which is considered a gold standard for pre-clinical development. ONC is a severe injury that is more reminiscent of a traumatic ocular injury (rather than glaucoma), performing ONC for these therapies would not provide additional evidence in the context of glaucoma. Please also see the response to Reviewer #1, comment #1 regarding testing novel compounds in *in vivo* models.

Assessing RGC density by counting consistent ROIs remains commonplace in the field (Bui group 2017 PMID: 28239332, Di Polo group 2022 PMID: 36103832, Calkins group 2022 PMID: 34984584, Inman group 2022 PMID:32422282, Martin group 2018 PMID: 30258047, Gordon group 2015 PMID: 26277582, amongst others). Counting ROIs has previously been demonstrated to provide solid estimation of true RGC density (PMID: 18241929, 25343338). While our counts may provide an underestimation of true RGC density, they are performed in a consistent manner as evidenced by low SD of our control groups. Whole retina counts are also performed by a number of groups, but these are predominantly performed in smaller scale experiments owing to the considerable time required to count all RGCs in a retina. We have assessed RGC density in over 450 retinas in this manuscript, whole retina counts on this scale would be unprecedented and completely impractical.

4. The effect of FK866 is only partial. Would one not expect that the effect of EGCG to be absolutely dependent on NMN? This may be the case as FK866 may also only decrease NMN by 30% (in which case there is absolute dependence on NMN), but this would need to be demonstrated.

Please see the response in Reviewer 2, comment #2 which also addresses this point.

5. There also are conflicting data on how levels of NMNAT2 and ECGC interact. It should be the case that both NMNAT2 levels and ECGC cooperate at low and intermediate levels of NMNAT2 but that once NMNAT2 levels are maximal in terms of NAD production, that ECGC no longer has an effect. This is inferred but not demonstrated.

The Authors agree that there is a maximal capacity for NAD production. Across multiple methods of raising cellular NAD levels in neural tissue (including via *Nmnat1* gene therapy, Williams et al., 2017 PMID: 28209901; *Wld^S* transgene, Williams et al., 2017 PMID: 28487632; high dose nicotinamide, Williams et al., 2017 PMID: 28209901, Tribble et al., 2021 PMID: 33932867) we have consistently reached a threshold/ceiling of ~200% of control NAD *in vivo*. Importantly this ceiling cannot be overcome (at least in the retinal and cortical neurons) when both additional enzyme and substrate are provided (*i.e.* *Wld^S* + NAM does not provide significantly elevated NAD by either alone; Figure 2 of PMID: 28487632). ECGC already reaches this threshold/ceiling of ~200% in neurons with basal levels of NMNAT2.

We have included the following in the Discussion to address this.

“Across multiple methods of raising cellular NAD levels in neural tissue (including via Nmnat1 gene therapy (3), Wld^S transgene (27), and high dose nicotinamide (3, 20)) we have consistently reached a threshold/ceiling of ~200% of control NAD (which is sufficient for strong/complete neuroprotection in the retina and optic nerve). ECGC at μM doses can achieve this level of NAD increase in neuronal tissue to a therapeutic level.”

6. Since NMNAT2 is present in other neurons in the CNS, and likely other cells too, the data on NMNAT2 dosage affecting RGC survival during development, aging and after injury are somewhat oversold; the possibility of brain or even systemic effects need to be acknowledged.

We agree that retinal ganglion cell survival during development and aging in *Nmnat2^{gtBay/gtE}* mice likely is influenced by brain connectivity. Developmentally-regulated retinal ganglion cell death is influenced by the ability of retinal ganglion cells to make connections in the SC, dLGN, and other target areas while injury to these areas can also influence RGC survival (PMID: 3782500; PMID: 30655352, PMID: 1577124, PMID: 1484120). Given that neurons in these brain regions will also have reduced *Nmnat2*, there is likely to be fewer supporting neurons or a more limited capacity to maintain retinal ganglion cell connectivity. We have amended the manuscript to reflect this nuance:

Results:

“Developmentally regulated RGC death and survival is influenced by connectivity in target brain regions (16, 17). Given that neurons in these brain regions will also have reduced Nmnat2, there is likely to be a more limited capacity to maintain RGC connectivity, which may also partially explain the developmental and early onset degeneration of RGCs in these mice.”

However, our injury model in *Nmnat2^{gtBay/gtE}* removed these influences. In the retinal explant model, retinas are removed from the eye and maintained in tissue culture and are therefore free from brain and systemic effects.

7. The description of the derivatization of ECGC is hard to follow. For example, why was compound 18 the basis for further derivatization when there were more potent variables

available. It also seems that all these details presented as results are really methods and not results, and that the actual results (how these new compounds behave in enzymatic vs neuroprotective assays) are not shown.

We have clarified the description of the derivation including an explanation for the importance of selecting Compound 18 as the basis for structural expansion. Since the drug development was guided by results in an iterative process, we feel this is important to include within the results section.

The following has been added to the Results:

“This led to the identification of Compound 18 as the simplest structure that showed statistically significant activity, which represented a promising drug-like hit for lead optimization. Analogues of Compound 18 were then tested to explore the structure–activity relationships around the chromene core. The synthetic accessibility of this particular chemical scaffold allowed for the preparation of a series of drug-like derivatives, leading to the identification of more potent compounds very rapidly. The removal of metabolically labile hydroxyl groups led to a decrease in activity, but their replacement with fluorine groups on the aromatic ring restored the NAD boosting activity, potentially increasing the lipophilicity and metabolic stability of the compounds (e.g. Compound 17, 20, 50, Figure 3C-D, Supp Fig 9). A scaffold hopping strategy led to the identification of three new cores: tetrahydroquinoline (Group 2), benzomorpholine (Group 3), and tetrahydroquinoxaline (Group 5), which showed a higher potency compared to the chromene core and were selected as scaffolds for the design of a new series of analogues. Across these iterations we identified several small molecules with greater potency in increasing NAD than EGCG.”

8. Not showing the actual chemical formulas of all the derivatives makes the science hard to evaluate, and makes the study impossible to reproduce. While IP makes sharing such data difficult, maybe the same conclusions could be made without disclosing the lead compounds (but disclosing others).

All structures were made available both as drawings and as SMILES in the original supplementary datasets. To add to this, the top 10 NAD producing compounds have been given an additional figure to highlight this (**Supplementary Figure 10**).

REVIEWER COMMENTS

Reviewer #1 (Remarks to the Author):

The authors have done a good job in providing new data or more explanation to address all the comments. This manuscript can now be accepted for publication.

Reviewer #2 (Remarks to the Author):

My concerns and comments have been sufficiently addressed by this revision. The authors have done an admirable job of provided detailed and thoughtful responses, as well as additional data, to answer my prior queries.

Reviewer #3 (Remarks to the Author):

In their revised version of the manuscript entitled "NMNAT2 is a druggable target to drive neuronal NAD production", Tribble et al. provide a rebuttal for the issues raised by this and the other reviewers. In the opinion of this reviewer, these responses are insufficient to assuage the concerns raised. While below are provided my remaining concerns, some raised by the other reviewers and fueled by the authors' responses, I want to reiterate that I find the whole line of investigation extremely exciting as I think that acting at the level of NMNAT2 could be transformative and circumvent some of the issues surrounding targeting the NAD synthetic/catabolic pathway by other means, as very eloquently outlined in the new discussion. So, I really want this to work, but I find the data so far unconvincing, and more problematically, I find an attempt to gloss over real concerns in order to sell this as a therapeutic approach, which the authors have an obvious personal interest in doing.

The concern raised before that most of the data presented here have been previously shown remains. The expression data showing NMNAT2 higher in RGCs than other retinal neurons was demonstrated by the same group in PMID:36681854. I presume the data shown here is not the same data as shown there, which would not be appropriate. Moreover, in that study there was an attempt to demonstrate by immunohistochemistry that the protein expression too was highest in RGCs, which would be far more convincing than relying on single cell and single nuclei transcriptomics, which have significant technical limitations that complicate their use as a quantitative tool. The results section begins with referencing AD studies that quantified protein expression for NMNAT2 by Westerns; why similar studies are not done here (e.g., to look at the NMNAT2 mouse allelic series) is not clear. An antibody exists. NMNAT2 is mainly RGC expressed. So, a whole retina Western should suffice. The immunos presented in PMID:36681854 were not particularly convincing, and the quantification methods used were non-standard and somewhat suspect. Maybe that explains why similar studies are not presented here. All in all, while I think the data does support that NMNAT2 is higher in RGCs than other retinal neurons, and in neurons higher than non-neuronal cells, the data are all supportive but not conclusive. And the fact that the more mechanistic studies are done in cortical cells, or as opposed to RGCs, raises some bells. That NMNAT2 is relatively specific to RGCs (and some other neurons) is fundamental to the development and use of pharmacological interventions centered at NMNAT2. Thus, this point really needs to be nailed. The team has the required tools to do so, including mice without NMNAT2, which are the essential tool to validate antibody reagents. Why such obvious studies are not conducted is somewhat perplexing.

Similarly, the allelic series effect on RGC viability is also not novel, as the Coleman group has already demonstrated this. More problematically, since there is a developmental effect on RGC numbers, and the referenced dendritic changes, the logic that links retinas with less NMNAT2 as being more vulnerable to NMNAT2 levels conferring such risk is inherently flawed. Yes, of course sickly retinas are going to be more susceptible to injuries such as putting them in culture, especially if developmentally RGCs never were generated in the right numbers, with all the likely sequelae (differences in circuitry, vascular development, etc.).

The entire figure 1, attempting to correlate NMNAT2 levels to susceptibility, is only correlative and

not particularly convincing. Finding that NMNAT2 varies more than some RGC markers seems like cherry picking. In a rigorous study of all genes, does NMNAT2 vary more than other genes, and does that variability explain susceptibility? Does NMNAT2 level in the context of the DBX strains correlate with susceptibility? The NMNAT2 allelic series is used to argue that NMNAT2 is important to RGCs because RGC number does correlate with NMNAT2 levels, and yet the exact opposite argument is made when it comes to DBX strains. The prediction, I think, is that NMNAT2 levels should be linked to susceptibility to axon injury, say by correlating NMNAT2 levels to survival after an optic nerve crush paradigm. If I recall those experiments, DBA/2J mice were particularly resistant to optic nerve crush injury. Do DBA/2J stand out as having more NMNAT2 than the more vulnerable strains? While I am sure one could come up with logic to explain away why NMNAT2 levels may matter in some contexts but not others, that is the whole problem with Figure 1, namely that it is subject to interpretation. The whole figure 1 is just correlative, has some inherent limitations such as not validating at the level of protein or enzymatic activities (i.e., RGC NAD levels), and is largely just reproducing data shown before. For this reason, it is not at all clear why it should be a primary figure rather than supplemental. The novelty of the current study is the Epigallocatechin gallate (ECGC) results, and the derivation of more potent agents. And, yet, much of those data are relegated to supplementary, such as the rotenone and JC1 experiments (though maybe because authors recently published a separate study using rotenone).

One of the main strengths of the study is the use of the NAMPT inhibitor FK866 to show that ECGC is presumably acting by regulating NMNAT2 itself. Upon further thought, however, these experiments also are potentially problematic. As mentioned in the rebuttal, inhibiting NAMPT is likely to affect NMNAT1 too, which would be predicted to affect photoreceptors, and therefore also Muller cells. So, how do they know that the interaction they observe in their whole retina models is due to the salvage pathway within RGCs? That experiment can be done in cultured cells, preferably RGCs. Extrapolating to the whole retina system is problematic. There is also a lack of clarity as to how these experiments were conducted. Among some of the details missing are when and how where the drugs added relative to one another, where doses used within physiological levels (e.g., non-saturating and still specific), what solvent controls were used. In order to interpret these crucial experiments, a very large dose of trust and inference is required. And yet, these are the most critical experiments.

While it is true that many labs use counts of Rbpm in selected fields to assess RGC viability, as opposed to retina wholemounts, that does not make the practice correct, especially when there is not sufficient detail provided as to how representative fields are selected. The variance in the counts would make one think that nearly the same regions of retina were selected for all counts. That would be hard to do without careful consideration of axes during dissection and that similar eccentricities were consistently being sampled (and of course, that investigators were blinded to the groups when retina fields were being selected for counting). This lab performs rigorous science, so I am sure all proper procedures were followed; however, detail still needs to be provided so that others can reproduce the findings.

In thinking further about the whole project, why is it that the study focusses on neuroprotection. Yes, the lab has demonstrated that, somewhat surprisingly, NAD levels affect RGC survival, and not just axon preservation. However, the primary effect of NAD levels is on axon preservation. So, why are they not using an axon, or even neurite metric, as their primary readouts to assess their ECGC and derivatized compounds? The Coleman group has elegant assays in place. So, why are they not done? The prediction is clear: if ECGC and related compounds are acting at the level of NMNAT2, they should phenocopy WldS and Sarm1 based interventions in those systems. What am I missing here?

As alluded to in response to the previous reviews, the most critical data is the effectiveness of ECGC in the OHT model. Unfortunately, those studies showed small effects, involved pre-treatment, and used field sampling, rather than either whole retina counts, or even better yet axon counts. While it is true that there are good reasons why such studies would be difficult and time consuming, the authors are wanting to publish their studies in a high profile journal and to make large claims. High profile journals and large claims carry with them associated large expectations. Yes, maybe the explant model might be sufficient, as other interventions (e.g., Sarm1 pharmacological interventions) were sufficiently exciting to be published in general readership

journals when only demonstrated in culture systems. However, because of the caveats mentioned above, this dying explant system might not enable the marriage of mechanistic and therapeutic studies the way that simpler culture models do, and is open to alternative explanations and possible misinterpretation.

I was also confused as to why they see NMNAT2 in the optic nerve head. Do they think the mRNA is axonal? Is there any evidence for that? This really needs to be explained clearly, as it may be evidence of non-RGC NMNAT2, which would obviously confound interpretation.

There is also now a concern that all the other postulated actions of ECGC (epigenetics, etc..), as claimed by others, are not being ruled out as possible mechanisms or at least possible confounders. While the FK866 studies do somewhat support the view that ECGC acts in the manner that they claim, these studies are not conclusive for the reasons stated above.

In order to make the strong claims that they do and reach the very high threshold of general interest, the authors really need to focus on their new drugs. While it is true that highly mechanistic definitive studies (marrying biophysics and enzymatics) may be too much to be expected at this point, expecting the authors to show that their drugs do two things (increase NAD in RGCs, and be neuro- or axon-protective) using the very same experimental paradigm (be that cultured RGCs or retina explants), is not too much to expect. There are sensitive NAD assays that maybe could be used on FACs or immune-panned RGCs. There are also some transgenic reporters that might be usable in vivo. Without definitive linking of the action of these drugs on NAD levels in RGCs to their neuro/axo-protective activities, the studies will remain correlative and potentially misleading.

R#3's comments in **bold**, Author's comments in plain text, quotes to the revised manuscript in "*quoted italics*".

In their revised version of the manuscript entitled "NMNAT2 is a druggable target to drive neuronal NAD production", Tribble et al. provide a rebuttal for the issues raised by this and the other reviewers. In the opinion of this reviewer, these responses are insufficient to assuage the concerns raised. While below are provided my remaining concerns, some raised by the other reviewers and fueled by the authors' responses, I want to reiterate that I find the whole line of investigation extremely exciting as I think that acting at the level of NMNAT2 could be transformative and circumvent some of the issues surrounding targeting the NAD synthetic/catabolic pathway by other means, as very eloquently outlined in the new discussion. So, I really want this to work, but I find the data so far unconvincing, and more problematically, I find an attempt to gloss over real concerns in order to sell this as a therapeutic approach, which the authors have an obvious personal interest in doing.

The Authors would like to thank R#3 again for their comments aiming to improve the manuscript further. We have done our best to address what we can, within the constraints of the limited resources available.

The Authors have been careful to comment that these novel compounds have potential but are not ready for clinical use at this stage.

To address this we have made the following comments within the text:

Abstract:

"This is the first class of NMNAT2 targeted small molecules and could have an important therapeutic impact for neurodegenerative disease following further drug development."

Introduction:

"...we develop the first small molecules driving neuronal NAD production through NMNAT2 which can also provide neuroprotection against RGC injury ex vivo."

Discussion:

"The novel small molecules developed here serve as a starting point for further medicinal chemistry to develop NMNAT2-targeted drugs that could be used in vivo, and potentially progress into human clinical trials. The next important steps will be further drug development focusing on ADME and toxicity in rodent and human cells lines, toxicity in vivo, and in vivo pharmacodynamics and pharmacokinetics to establish dose/response ranges which can then be tested in mature, in vivo, animal models of glaucoma with multiple disease metric assessed (i.e. visual function, optic nerve and axon morphologies, mitochondrial dynamics). As NMNAT2 has been implicated as a key component of axon degeneration, if properly tested, these compounds have potential for other neurodegenerative diseases."

The concern raised before that most of the data presented here have been previously shown remains. The expression data showing NMNAT2 higher in RGCs than other retinal neurons was demonstrated by the same group in PMID:36681854. I presume the data shown here is not the same data as shown there, which would not be appropriate. Moreover, in that study there was an attempt to demonstrate by immunohistochemistry that the protein expression too was highest in RGCs, which would be far more convincing that relying of single cell and single nuclei transcriptomics, which have significant technical limitations that complicate their use as a quantitative tool.

This is indeed re-use of published, publicly available data from other labs as well as our own.

No attempts have been made to claim this as novel (so our sincere apologies if this has been seen as such). A lot of the data in Figure 1 is re-analysis of publicly available datasets. This is detailed in the methods as well as being the second line of the results section:

“To explore NMNAT2 in the retina we first queried publicly available ...”

To prevent any miscommunication this is now also stated in the figure legend to prevent any miscommunication if a reader were to be scanning the figures:

“Data in A-F was generated through screening publicly available datasets (see Methods).”

We previously demonstrated expression of *Nmnat2* and other NAD-pathway enzymes across all retinal cells. We reproduce this in the left panel (importantly not the same plot as previously reported) of E to highlight retinal ganglion cell specificity but the data in the middle and left panel is a new analysis of individual retinal ganglion cell expression values.

In the current version of the manuscript we have retained these panels in E and have called out directly in the figure legend (in addition to the methods and results). As the scientific community is generating more data than it can possibly use or analyse, we believe that re-analysis of public datasets is a perfect way to build hypotheses which can then be empirically tested in-house. This has many advantages over re-generating the same data for personal use including reduced costs, time, and ethical considerations.

One request made is documentation that NMNAT2 is specific / enriched in retinal ganglion cells. This is what we have shown in Figure 1. As such (and given that *Nat Comms* is online only) we believe this can remain as an up-front figure and not buried in the supplementary data. The Authors believe this is important as it shows how these hypotheses are generated. We agree with R#3 that *more* data should be shown upfront in the main figures and we have reordered the new figures (from 3 main figures and 11 supplemental to 8 main figures and 8 supplemental).

To add to these points, in the first round of review R#3 specifically commented that nuRNA-seq is not sufficient, so we supplemented with scRNA-seq data to add this and to reduce concerns. These new data analyses validate the previous findings and supplement the data in Figure 1E.

Regarding IHC staining of retina sections, R#3 makes a valid point that protein quantification would perfectly supplement the genetic data presented here. We have previously shown NMNAT2 protein by IHC in human retina sections. As pointed out in a later comment, these methods are only semi-quantitative and are not strictly retinal ganglion cell specific in the IPL as there is a mosaic of neurites from other retinal ganglion cells and amacrine cells which express NMNAT2. However, there are no well-validated antibodies for NMNAT2 via WB that work across mouse, rat, and human tissues. Even in these cases, isolation of individual retinal ganglion cells would be impossible from FFPE tissue and would fully consume clinical samples from the biobank (which we do not have the ability to do).

As we agree that protein level in single retinal ganglion cells would be a perfect dataset to have (but is caveated by significant technological hurdles) we have added the following to the discussion:

“However, NMNAT2 protein levels in individual RGCs remains to be definitively assessed ...”

The results section begins with referencing AD studies that quantified protein expression for NMNAT2 by Westerns; why similar studies are not done here (e.g., to look at the NMNAT2 mouse allelic series) is not clear. An antibody exists. NMNAT2 is mainly RGC expressed. So, a whole retina Western should suffice.

As stated above, although we agree with R#3 that protein levels of NMNAT2 would be ideal there are a few technological hurdles with the approaches made in these papers:

1. In the AD paper, the authors use large volumes of human tissue with a human specific antibody (this is not relevant to our work in mouse retinas [billions of neurons vs. thousands of neurons]) or used transfected cell lines to generate sufficient quantities of NMNAT2 protein for protein assays,
2. The WB antibodies for rodent NMNAT2 are not specific and should not be used for gross quantification of NMNAT2,
3. We do not have control post-mortem human retina tissue that is free from neurodegeneration (only FFPE pathology tissue or retina tissue from the cornea biobank which is 48-72 h post mortem and therefore will already be undergoing a loss of NMNAT2 and Wallerian degeneration).

The immunos presented in PMID:36681854 were not particularly convincing, and the quantification methods used were non-standard and somewhat suspect. Maybe that explains why similar studies are not presented here.

Since in 36681854 we used FFPE sectioned tissue from a biobanked archive, antibody labelling is the only available tool to assess NMNAT2 levels in this very rare, very valuable tissue. As the antibodies are not 100% specific, we opted against using these in mouse, rat, or additional tissues in the current manuscript. Please also see the comments made above and the added text to the discussion:

“However, NMNAT2 protein levels in individual RGCs remains to be definitively assessed which will require the development of RGC specific tools to label NMNAT2 or highly specific antibodies to NMNAT2 suitable for protein quantification.”

All in all, while I think the data does support that NMNAT2 is higher in RGCs than other retinal neurons, and in neurons higher than non-neuronal cells, the data are all supportive but not conclusive. And the fact that the more mechanistic studies are done in cortical cells, or as opposed to RGCs, raises some bells.

The Authors agree that experiments using only retinal ganglion cells would add valuable data. However, it is worth pointing out that cortical neurons were used to develop an NMNAT2 targeted compound whereas retinal systems were used to assess neuroprotection in a ‘glaucoma-like’ setting (our primary indication given the available information and the skill set of our lab). As the drug development work requires large volumes of neurons in suspension (the primary assay – NAD luminometry) we opted to use cortical neurons isolated direct from murine cortex as opposed to generating retinal ganglion cells *en masse*.

Tools to use retinal ganglions *en masse* are highly caveated by either:

- a. These are immuno-panned or FAC-sorted cells and as such are devoid of dendrites and axons (major sites of NMNAT2 protein) unless subsequently grown with neurotrophic factors (which will no doubt change retinal ganglion cell make up), or,
- b. Grown *en masse* from iPSCs which do not retain epigenetic memory or represent adult, in-circuit, retinal ganglion cells.

As culturing/isolating retinal ganglion cells is very low yield (*i.e.* a mouse has ~50,000 retinal ganglion cells, so 50% of a single well of a 24 well plate at a 100% retrieval rate) we opted to first have a test bed of cortical neurons which we can mass produce in the 100s of millions. (*N.B.* for the primary assay each well contains 100,000 cortical cells.) This has been made clear in the methods and results:

“Cortical, spleen, liver, and muscle samples were diluted to a concentration of 2 million cells/mL with HBSS (in which 100,000 cells were used per assay well).”

“(Due to the low yield (>50,000 RGCs/retina), NAD levels in purified RGCs were not assessed in these studies.)”

Lastly, it is important to note that cortical neurons were not used in any instance to make claims about retinal ganglion cells, their survival potential, or their NMNAT2 levels. This has been made clear through the manuscript.

That NMNAT2 is relatively specific to RGCs (and some other neurons) is fundamental to the development and use of pharmacological interventions centered at NMNAT2.

Glaucoma is a common neurodegenerative disease and has been demonstrated to have similarities to other neurodegenerative diseases. We use glaucoma here as a ‘tool disease’. In the manuscript we do not claim that NMNAT2 is specific to *only* retinal ganglion cells in the body, nor even the retina. (Or that our novel compounds are specifically targeting retinal ganglion cells (although it would be predicted that they would be the primary affected cell type).)

NMNAT2 has been demonstrated in almost all neurodegenerative diseases. NMNAT2 is expressed in all assessed neuronal cells and therefore is a valid target for many indications. The following has been added to the manuscript:

“Increasing levels of NAD provides neuroprotection in multiple cell and animal models of disease and in human clinical trials (1).”

“Our study demonstrates that NMNAT2 expression is overwhelmingly restricted to RGCs in the retina (although it is present in the majority of neuronal cell types in the body ...”

“Novel small molecules that target NMNAT2 could provide neuronal targeted increases in NAD to boost NAD pools and prevent the initiation of neurodegenerative cascades. This could be relevant in many neurodegenerative diseases in which a depletion of NAD generating capacity or NMNAT2 levels have been demonstrated.”

To add to these points, even if NMNAT2 is expressed in other retinal cell types there is no reason to expect this to be a problem (as there seems to be no toxicity in having ‘high’ NAD levels based on the ~80 years of clinical history for NAD-related supplements [11126400] and that pan-neuronal NMNAT2 overexpression does not have deleterious effects [23995269]). The high expression at transcript level in retinal ganglion cells does suggest it has a particularly important role for these cells. This is supported by the fact that retinal ganglion cells are particularly sensitive to fluctuations in NAD generating capacity and that nicotinamide can prevent retinal ganglion cell injury in many models.

Thus, this point really needs to be nailed. The team has the required tools to do so, including mice without NMNAT2, which are the essential tool to validate antibody reagents. Why such obvious studies are not conducted is somewhat perplexing.

It is important to comment here that there are no mice without NMNAT2 as NMNAT2 complete KO is lethal. In this manuscript we demonstrate that compound heterozygous mice have 25% NMNAT2 and therefore could not be used to validate an antibody. Generating a KO line and developing a new antibody is outside of the scope of a rebuttal / review.

The following has been added to the discussion as a future direction:

“Supporting that this occurs due to specificity for NMNAT2, EGCG’s NAD boosting and neuroprotective effects are reduced or blocked in the presence of the NAMPT inhibitor FK866 or with Nmnat2 depletion (it is important to note that in these Nmnat2 depleted systems, ~25%”

Nmnat2 activity remains. As complete Nmnat2 KO mice are not viable, further development of conditional tools of Nmnat2 would be of benefit to further confirm these findings)."

Similarly, the allelic series effect on RGC viability is also not novel, as the Coleman group has already demonstrated this.

The mice are not novel (nor do we claimed to have made them). The following is clarified in the methods:

"Nmnat2 gene-trap allele mouse lines Nmnat2^{+/gtBay} and Nmnat2^{+/gtE} (as previously described (15)) were kindly provided by Michael P. Coleman."

However, the allelic series in the retina and their NMNAT2 levels have not been previously published. (*i.e.* this is novel data.)

More problematically, since there is a developmental effect on RGC numbers, and the referenced dendritic changes, the logic that links retinas with less NMNAT2 as being more vulnerable to NMNAT2 levels conferring such risk is inherently flawed. Yes, of course sickly retinas are going to be more susceptible to injuries such as putting them in culture, especially if developmentally RGCs never were generated in the right numbers, with all the likely sequelae (differences in circuitry, vascular development, etc..).

Regarding the "sickly" retina, we have not quoted this anywhere in the manuscript, but we agree that given our findings it could be an avenue for further research (*i.e.* consequences of low NMNAT2 during retinal development). This is a separate line of experimental inquiry and we have added the following to the results:

"Low Nmnat2 (Nmnat2^{gtBay/gtE}) results in a developmental or very early post-natal reduction in RGC density followed by an age-related loss of RGCs initiated between 6 and 12 months of age which is absent in Nmnat2^{+/+} mice up to 22 months of age (Figure 1I, Supp Fig 2A) (although further experiments are required to elucidate the exact timing of this neuronal reduction).

Importantly when we assessed neurodegeneration our controls are both uninjured mice of the same genotype as well as wildtype controls in order to control for other possible existing effects.

Regarding the comments about further sequelae, for these experiments the retinas are maintained *ex vivo* in tissue culture and this removes the influence of vasculature and circuitry. As there is no photoreceptor stimulation this removes any potential caveat in changes to retinal circuitry.

The entire figure 1, attempting to correlate NMNAT2 levels to susceptibility, is only correlative and not particularly convincing. Finding that NMNAT2 varies more than some RGC markers seems like cherry picking.

This data was added by request in the previous review by R#3 and currently supports the narrative in the manuscript.

In a rigorous study of all genes, does NMNAT2 vary more than other genes, and does that variability explain susceptibility? Does NMNAT2 level in the context of the DBX strains correlate with susceptibility?

We do not have the opportunity to screen neurodegenerative insults across the BXD lines as we do have this colony (as we used publicly available genetic data as outlined clearly in methods).

The NMNAT2 allelic series is used to argue that NMNAT2 is important to RGCs because RGC number does correlate with NMNAT2 levels, and yet the exact opposite argument is made when it comes to DBX strains.

In the BXD lines we are exploring fluctuations within a normal physiological range (*i.e.* normal changes in genetic variation); therefore, we would not have the sensitivity to assess retinal ganglion cell numbers vs. NMNAT2 levels in these mice

In the allelic series we are exploring a pathophysiological role for damagingly low NMNAT2. These are very separate experiments. The following has been added to the results to clarify this:

“We demonstrate that across recombinant inbred BXD strains (recombinant inbred lines from crosses between C57BL/6J mice (B6) and DBA/2J mice (D2) to explore Nmnat2 variations within a non-pathological range), Nmnat2 expression is highly variable (up to 2-fold difference) among individual strains and is variable within independent data sets for whole eye, retina, and midbrain (where RGC axons terminate; Figure 1A) suggesting variability in expression within the whole visual system.”

The prediction, I think, is that NMNAT2 levels should be linked to susceptibility to axon injury, say by correlating NMNAT2 levels to survival after an optic nerve crush paradigm.

We cannot assess NMNAT2 levels *and* injury in the same tissue, as both processes require the tissue to be consumed for assay. Instead, we have used mice with known levels of NMNAT2 (*i.e.* the allelic series) and perform an axon injury (in our case axotomy which we chose over optic nerve crush, as the axotomy injury comes without caveats driven by mass inflammation *etc.*). Therefore, an optic nerve crush would be a reductive experiment as this has already been shown. We agree that validation of low / high NMNAT2 in other axon injuries would further strengthen the support for NMNAT2 in neurodegenerative disease, as such the following has been added the discussion:

“We demonstrate that reduced Nmnat2 expression drives susceptibility to axon injury (axotomy) and future experiments could assess this long term in more chronic optic nerve injury models in these mice (e.g. optic nerve crush at different ages). Supporting these findings, restoring NMNAT2 expression in depleted retinas via gene therapy removes this susceptibility to RGC degeneration and provides neuroprotection. This neuroprotection is maintained through to a chronic in vivo glaucoma model where the protection is not fully complete likely due to the other OHT-related events (i.e. changes in vascular tone, neuroinflammation) which a neuron-specific therapy cannot overcome.”

If I recall those experiments, DBA/2J mice were particularly resistant to optic nerve crush injury.

In the manuscript that studied this there is an 8% difference in cell survival, not “resistance” *per se*. Importantly, this 8% difference is the difference in the number of retinal ganglion cells in control (*i.e.* normal numbers of cells), whereas both strains reach the same end stage as shown below in the figure below copied from the paper. As these changes are small, and as the tissue reaches the same end stage, we do not feel that this warrants further discussion in the manuscript.

Figure 2

Do DBA/2J stand out as having more NMNAT2 than the more vulnerable strains?

In the BXD series (Figure 1) both B6 and D2 are represented and neither of these strains have the most upper or lower values (D2 = 10.69, B6 = 11.09, min for series = 10.23, max for series = 11.29 RMA gene level). This data has been added to the figure legend:

“(B) In the retina, this variability is not related to the number of RGCs (Spearman's rank correlation) and the variance in Nmnat2 expression is significantly greater than for RGC markers Pou4f1, Rbpms, and Tubb3. Both the founder strains (B6 and D2) are represented in these data and neither of these strains have the most upper or lower values (D2 = 10.69, B6 = 11.09, min for series = 10.23, max for series = 11.29 RMA gene level).”

While I am sure one could come up with logic to explain away why NMNAT2 levels may matter in some contexts but not others, that is the whole problem with Figure 1, namely that it is subject to interpretation. The whole figure 1 is just correlative, has some inherent limitations such as not validating at the level of protein or enzymatic activities (i.e., RGC NAD levels), and is largely just reproducing data shown before.

As before, we have been clear about use of data, and we will endeavour to make sure that this is all clear throughout the manuscript (please see the points above). We do not have the ability to recover 100s of BXD strains (i.e. 1000s of mice) to assess NMNAT2 enzymatic levels (nor is there an assay for NMNAT2 enzymatic levels which is not an NAD assay). So, whilst we agree that these would be insightful experiments to conduct, they should be considered out of the scope of this current manuscript.

We have explained in what contexts and why NMNAT2 levels have an effect. Normal variation in NMNAT2 appear to carry no effect on retinal ganglion cell numbers (as has been shown in other neuronal systems) but very low levels enhance / drive susceptibility to neurodegeneration (in the case of the allelic series ~75% loss of NMNAT2 is sufficient). We have clarified this point throughout the manuscript. The following has been clarified in the results:

“Given that neurons in these brain regions will also have reduced Nmnat2, there is likely to be a more limited capacity to maintain RGC connectivity, which may also partially explain the developmental and early onset degeneration of RGCs in these mice. Even when accounting for this developmental drop out in RGC populations, RGC loss was significantly greater in Nmnat2^{gtBay/gtE} mice following axotomy (but not for the single alleles; i.e. predicted 75% and 50% Nmnat2) (Figure 1J, Supp Fig 2B). This suggests a threshold of Nmnat2 expression past which RGCs are sensitized to neurodegenerative insults (these thresholds are only met with genetic depletion and are not met within the normal variations within the population, Figure 1B). This is supported by the wide range of NMNAT2 levels in humans which are non-pathogenic in the absence of other neurodegenerative insults (e.g. NMNAT2 expression is

highly variable in aged postmortem human brains, and further decreased in brains with Alzheimer's disease (~75-100% of controls levels) (10)."

For this reason, it is not at all clear why it should be a primary figure rather than supplemental.

This is point for the editorial and copyediting team to address. As *Nat Comms* is online only we would argue for inclusion as a main figure. We feel that the genetic data is important to the narrative and the rationale of the hypothesis.

The novelty of the current study is the Epigallocatechin gallate (EGCG) results, and the derivation of more potent agents.

We agree.

And, yet, much of those data are relegated to supplementary, such as the rotenone and JC1 experiments (though maybe because authors recently published a separate study using rotenone).

Please see our comment above regarding reordering of figures. The rotenone studies are now in the main manuscript. It is worth noting however that the rotenone study was performed, written, submitted, reviewed, and accepted in the time that this manuscript has been out at review. In addition, this rotenone paper did not study NMNAT2 or EGCG (it is a paper on mitochondrial morphology during mitochondrial stress in the retina).

One of the main strengths of the study is the use of the NAMPT inhibitor FK866 to show that EGCG is presumable acting by regulating NMNAT2 itself. Upon further thought, however, these experiments also are potentially problematic. As mentioned in the rebuttal, inhibiting NAMPT is likely to affect NMNAT1 too, which would be predicted to affect photoreceptors, and therefore also Muller cells. So, how do they know that the interaction they observe in their whole retina models is due to the salvage pathway within RGCs?

NMNAT1 and NMNAT2 are both terminal enzymes for the salvage pathway. NMNAT1 is present in all nucleated cells, including retinal ganglion cells. If NMN production is inhibited, this will of course lower the output of both NMNAT1 and 2. In the data presented, FK866 doesn't completely *remove* NMN (this was a major point in the previous rebuttal) as: 1. There will still be residual NMN prior to inhibition of NAMPT which has yet to be consumed, and 2. There is still activity of NMNAT1. As we see an effect in these systems, it is suggestive, but not conclusive that these effects must come from NMNAT2. We further supplement this data using the allelic series, where mice with only 25% NMNAT2 no longer show neuroprotection. It is important to note that at no point do we comment that FK866's actions on NAMPT, or NMN, are specific to NMNAT2. Regarding blocking NMN production in other cells in the retina, it is important to note that FK866 alone had no effect on retinal ganglion cell survival in culture. This has been clarified in the results:

"We depleted NMN (the substrate for NMNAT2) using FK866 (a specific NAMPT inhibitor, the upstream enzyme to NMNAT1 and NMNAT2)."

and

"This demonstrates that EGCG requires NMN to generate NAD and to provide neuroprotection (i.e. an NAD-salvage pathway dependent process requiring NMNAT1 or NMNAT2 to further stimulate NAD production)."

That experiment can be done in cultured cells, preferably RGCs.

Please see the prior points re: experiments in retinal ganglion cells.

There is also a lack of clarity as to how these experiments were conducted. Among some of the details missing are when and how where the drugs added relative to one another, where doses used within physiological levels (e.g., non-saturating and still specific), what solvent controls were used. In order to interpret these crucial experiments, a very large dose of trust and inference is required. And yet, these are the most critical experiments.

These points have been double checked and clarified in the methods.

While it is true that many labs use counts of Rbpms in selected fields to assess RGC viability, as opposed to retina wholemounts, that does not make the practice correct, especially when there is not sufficient detail provided as to how representative fields are selected.

All our counts were performed on retinal wholemounts (aka flatmounts). This point was addressed in the previous rebuttal. In the methods the representative fields are clearly documented:

“Six images per retina were taken equidistant at 0, 2, 4, 6, 8, and 10 o’clock about a superior to inferior line through the optic nerve head (1000 μ m eccentricity). Images were cropped to 100 x 100 μ m and RBPMS+ cells were counted using the cell counter plugin for Fiji; counts were averaged across the 6 images.”

The variance in the counts would make one think that nearly the same regions of retina were selected for all counts.

This is correct, all images were taken at an eccentricity of 1000 μ m in a clockwise fashion. These data highlight that our method is reproducible.

That would be hard to do without careful consideration of axes during dissection and that similar eccentricities were consistently being sampled (and of course, that investigators were blinded to the groups when retina fields were being selected for counting).

The explant model was performed on mouse retinas. It is well documented that mouse retinas do not have a macular (region of dense retinal ganglion cells). As we are counting samples at 6 clockwise points (as above), and averaging there can be no bias due to location of sampling in the retina. As mouse retina have more dense retinal ganglion cells closer to the optic nerve head we controlled for eccentricity (*i.e.* always 1000 μ m from the optic nerve head). All the cell counts were blinded to treatment. This has been clarified in the methods:

“Cells were counted by a minimum of two observers blinded to treatment.”

This lab performs rigorous science, so I am sure all proper procedures were followed; however, detail still needs to be provided so that others can reproduce the findings.

We agree and strive to do reproducible science. Our recent work with NAD has been replicated not only by others in other animal systems but also independently in clinical trials.

These points have been double checked and clarified in the methods.

In thinking further about the whole project, why is it that the study focusses on neuroprotection.

Neuroprotection was the focus of this study as neuroprotective therapies for neurodegenerative disease are of great therapeutic need. We chose glaucoma as a primary indication for neuroprotection as 1. This is the primary focus of my lab, 2. There are no neuroprotective therapies for glaucoma.

Yes, the lab has demonstrated that, somewhat surprisingly, NAD levels affect RGC survival, and not just axon preservation.

Raising NAD with nicotinamide provides a robust soma protection in many different animal models. Given the wealth of prior studies on Wallerian degeneration in neuronal systems axonal death is intrinsically linked to cell survival unless other genetic determinants are there (e.g. in the context of glaucoma, *BAX*^{-/-} leads to soma survival but not axonal, whereas *SARM1*^{-/-} results in axon but not soma survival: 16103918 & 29526794). As mentioned in the previous rebuttal, this manuscript included over 450 retinas – to repeat these experiments to also collect optic nerves would not be feasible nor ethical. As optic nerve analysis would be an important next step when moving the drug development from *in vitro* and *ex vivo* work we have added the following to the discussion:

“The novel small molecules developed here serve as a starting point for further medicinal chemistry to develop NMNAT2-targeted drugs that could be used in vivo, and potentially progress into human clinical trials. The next important steps will be further drug development focusing on ADME and toxicity in rodent and human cells lines, toxicity in vivo, and in vivo pharmacodynamics and pharmacokinetics to establish dose/response ranges which can then be tested in mature, in vivo, animal models of glaucoma with multiple disease metric assessed (i.e. visual function, optic nerve and axon morphologies, mitochondrial dynamics). As NMNAT2 has been implicated as a key component of axon degeneration, if properly tested, these compounds have potential for other neurodegenerative diseases.”

However, the primary effect of NAD levels is on axon preservation. So, why are they not using an axon, or even neurite metric, as their primary readouts to assess their EGCG and derivatized compounds?

Raising NAD levels, e.g. by nicotinamide treatment, are protective to the soma. Our work here stems from our previous work that nicotinamide provides a robust somal protection in glaucoma models and improves visual function in glaucoma patients (28209901, 33932867, 32721104). As such, assessing the inner retina is a valid metric.

The Coleman group has elegant assays in place. So, why are they not done?

From this comment it is unclear to which assays R#3 is referring to. If R#3 alluding to the neurite degeneration assays, these are not feasible experiments for review: i.e. take all 58 conditions (56 compounds, EGCG, control) at 3 doses and 5 time points and perform these assays in DRG neurons as per Coleman’s previous work (i.e. 870 conditions with 3-6 replicates). It should also be noted that this would add no additional value to the retinal ganglion cell specific concerns raised by R#3.

The prediction is clear: if EGCG and related compounds are acting at the level of NMNAT2, they should phenocopy WldS and Sarm1 based interventions in those systems. What am I missing here?

R#3 is correct that EGCG, if raising NAD should provide neuroprotection. However, it is important to note that in retinal ganglion cell systems, SARM1 inhibition saves only the axon

and not the soma (29526794). In rats (but not mice), WLDS only protects the axon and not the soma (18783366).

As alluded to in response to the previous reviews, the most critical data is the effectiveness of EGCG in the OHT model. Unfortunately, those studies showed small effects, involved pre-treatment, and used field sampling, rather than either whole retina counts, or even better yet axon counts.

For the EGCG experiments, these are purely proof of concept as we do not envision that EGCG could become a drug (this was a major point of discussion in the prior rebuttal and revised manuscript). Therefore, its magnitude of effect in glaucoma is less important than whether there was an effect at all that could be followed up on. Importantly, and as made clear in the manuscript, EGCG has very poor bioavailability and is highly labile – therefore we did not hypothesize a large effect size. There is the purpose of the drug development project. Regarding pre-treatment, we also assess interventional treatment using the axotomy explant model in mouse and human retina. We have made sure to tone down any claims re: EGCG as a strong neuroprotective agent *in vivo*. The following has been edited in the results:

“EGCG provides full neuroprotection against RGC loss following axotomy mirroring the hNMNAT2 gene therapy (Figure 2B) and our previous work with nicotinamide (NAM; an upstream precursor to NAD in the salvage pathway) (3, 20). Further supporting this, EGCG delivered orally reduced (but not fully prevented) glaucomatous neurodegeneration in the rat OHT glaucoma model, with this neuroprotection improving with the addition of hNMNAT2 gene therapy (Figure 2C, Supp Figure 3D).”

While it is true that there are good reasons why such studies would be difficult and time consuming, the authors are wanting to publish their studies in a high profile journal and to make large claims. High profile journals and large claims carry with them associated large expectations. Yes, maybe the explant model might be sufficient, as other interventions (e.g., Sarm1 pharmacological interventions) were sufficiently exciting to be published in general readership journals when only demonstrated in culture systems. However, because of the caveats mentioned above, this dying explant system might not enable the marriage of mechanistic and therapeutic studies the way that simpler culture models do, and is open to alternative explanations and possible misinterpretation.

The explant model was used as it allows for a proof of concept (in which Wallerian degeneration is a key feature, and in which the *Wld^S* allele blocks soma death: 36635457) in which we do not have to fully develop a drug in terms of ADME, toxicity studies, regulatory documentation, and ethical approval. It exists purely as a proof of concept. We will make this clear throughout the manuscript. The following has been added to discussion to make clear the need for further *in vivo* testing:

*“The novel small molecules developed here serve as a starting point for further medicinal chemistry to develop NMNAT2-targeted drugs that could be used *in vivo*, and potentially progress into human clinical trials. The next important steps will be further drug development focusing on ADME and toxicity in rodent and human cells lines, toxicity *in vivo*, and *in vivo* pharmacodynamics and pharmacokinetics to establish dose/response ranges which can then be tested in mature, *in vivo*, animal models of glaucoma with multiple disease metric assessed (i.e. visual function, optic nerve and axon morphologies, mitochondrial dynamics). As NMNAT2 has been implicated as a key component of axon degeneration, if properly tested, these compounds have potential for other neurodegenerative diseases.”*

It is important to note that the explant experiment is only used for one mechanistic experiment which is FK866 blocking the effect of EGCG on cell survival. In all other instances it is used

as a model of neurodegeneration to test the role of NMNAT2 on neurodegeneration. Therefore, the fact that it is a 'dying' model is a feature for its use, not a caveat.

I was also confused as to why they see NMNAT2 in the optic nerve head. Do they think the mRNA is axonal? Is there any evidence for that? This really needs to be explained clearly, as it may be evidence of non-RGC NMNAT2, which would obviously confound interpretation.

Yes, this mRNA can be axonal (15473850), and there is protein translation in mature axons (32087477). We have demonstrated that NMNAT2 mRNA is low in non-neuronal cell types in the retina which is further validated across many human tissues in the human protein atlas (for which cell types in the retina are not annotated) (<https://www.proteinatlas.org/ENSG00000157064-NMNAT2>).

There is also now a concern that all the other postulated actions of ECGC (epigenetics, etc.), as claimed by others, are not being ruled out as possible mechanisms or at least possible confounders. While the FK866 studies do somewhat support the view that ECGC acts in the manner that they claim, these studies are not conclusive for the reasons stated above.

In the previous rebuttal other effects of ECGC were discussed in response to comments from reviewers. The following is in the discussion:

"As a polyphenol, ECGC has a number of potential mechanisms of action that could be providing neuroprotection including through modulation of reactive oxygen species, cell signalling, growth factors, autophagy, and apoptotic cascades (43). Modulation of these could also affect downstream NAD levels (44). Glaucoma is a complex neurodegenerative disease in which these factors have previously been identified as potential pathological mechanisms (45)."

In order to make the strong claims that they do and reach the very high threshold of general interest, the authors really need to focus on their new drugs. While it is true that highly mechanistic definitive studies (marrying biophysics and enzymatics) may be too much to be expected at this point, expecting the authors to show that their drugs do two things (increase NAD in RGCs, and be neuro- or axon-protective) using the very same experimental paradigm (be that cultured RGCs or retina explants), is not too much to expect. There are sensitive NAD assays that maybe could be used on FACs or immune-panned RGCs.

R#3 is correct and we have included neuroprotection data in Fig 3G.

Unfortunately, there are no available tools to spatially assess NAD purely in retinal ganglion cells. We cannot assess NAD in a retinal ganglion cell pure system given that tools do not exist. The only conceivable solution could be FACS sorting retinal ganglion cells, however this is highly caveated as:

1. Mice only have 50,000 retinal ganglion cells requiring a pooling strategy, and,
2. The method takes a long time leading to degraded or altered levels of NAD.

To assess NAD levels in retinal ganglion cells would introduce the caveats detailed above. These methods are also not conducive to the large numbers of cells required for NAD assays (in our hands 100,000 neurons per well) and the large number of conditions. These comments are addressed above.

There are also some transgenic reporters that might be usable in vivo.

There are no transgenic reports for NAD or NMNAT2. There is a NMNAT2-Venus mouse (currently only available as frozen stock), however, this transgene provides neuroprotection /

altered protein stability itself and therefore could not be used for these types of assays (24284888, 23995269, 23610559).

Without definitive linking of the action of these drugs on NAD levels in RGCs to their neuro/axo-protective activities, the studies will remain correlative and potentially misleading.

As mentioned above, the same retina cannot be used for both NAD levels and neuroprotection. We do not have the tools to assess NAD specifically in retinal ganglion cells (detailed previously). We have now addressed that defining retinal ganglion cell specific NAD metabolism will be a key next step in the understanding of retinal ganglion cell susceptibility to low NAD levels / low NMNAT2 levels in the discussion:

“Importantly, these top compounds were able to provide neuroprotection against RGC degeneration ex vivo, supporting their utility and potential as neuroprotective therapeutics in the future. Next steps could further define the effect of these compounds at a single cell level with specificity to RGCs. This would allow the elucidation as to whether only RGCs require increased NAD generating capacity for neuroprotection, or whether other neurons in the retinal circuitry contribute to this neuroprotection. These experiments could also assess whether sub-types of RGCs are more or less susceptible to low or high NMNAT2 levels.”

REVIEWERS' COMMENTS

Reviewer #3 (Remarks to the Author):

I thank the authors for providing such extensive arguments as to why the data stand. While in my view some of the conclusions remain somewhat of an overreach, the claims have been softened and the more obvious caveats have been made more explicit. In light of these improvements and the fact that the other two reviewers, who were thoughtful and thorough, agreed in thinking that the manuscript was sufficiently strong to merit publication, I too sign off and agree that the story is sufficiently strong to merit publication. Hopefully the authors and others will rush to do all the required mechanistic work to prove beyond any doubt that the drugs are acting as they think they are, and all the other hard work to determine whether Nmnat2-directed interventions are safe and effective and worth moving into humans.